# Distributed rhythm generators underlie *Caenorhabditis elegans* forward locomotion

**Anthony D Fouad[1], Shelly Teng[1], Julian R Mark[1], Alice Liu[1], Pilar Alvarez-Illera[1], Hongfei Ji[1], Angelica Du[1], Priya D Bhirgoo[1], Eli Cornblath[1], Sihui Asuka Guan[2], Christopher Fang-Yen[1,3]\***

[1]Department of Bioengineering, School of Engineering and Applied Science, University of Pennsylvania, Philadelphia, United States; [2]Lunenfeld-Tanenbaum Research Institute, Mount Sinai Hospital, Toronto, Canada; [3]Department of Neuroscience, Perelman School of Medicine, University of Pennsylvania, Philadelphia, United States

**Abstract** Coordinated rhythmic movements are ubiquitous in animal behavior. In many organisms, chains of neural oscillators underlie the generation of these rhythms. In *C. elegans*, locomotor wave generation has been poorly understood; in particular, it is unclear where in the circuit rhythms are generated, and whether there exists more than one such generator. We used optogenetic and ablation experiments to probe the nature of rhythm generation in the locomotor circuit. We found that multiple sections of forward locomotor circuitry are capable of independently generating rhythms. By perturbing different components of the motor circuit, we localize the source of secondary rhythms to cholinergic motor neurons in the midbody. Using rhythmic optogenetic perturbation, we demonstrate bidirectional entrainment of oscillations between different body regions. These results show that, as in many other vertebrates and invertebrates, the *C. elegans* motor circuit contains multiple oscillators that coordinate activity to generate behavior.

DOI: https://doi.org/10.7554/eLife.29913.001

**\*For correspondence:**
fangyen@seas.upenn.edu

**Competing interests:** The authors declare that no competing interests exist.

## Introduction

Oscillatory neural activity underlies rhythmic animal behaviors such as feeding and locomotion. Rhythm generating units are sometimes functional in isolated spinal cord and invertebrate nerve cord preparations, producing fictive rhythmic motor outputs that resemble in vivo patterns (*Marder and Calabrese, 1996*; *Marder et al., 2005*; *Kiehn, 2006*; *Mullins et al., 2011*; *Grillner and El Manira, 2015*). At the same time, sensory feedback and reflex loops have also been found to be important for motor rhythm coordination and modulation (*Wendler, 1974*; *Andersson et al., 1981*; *Yu and Friesen, 2004*; *Kristan et al., 2005*).

How do cellular pacemakers, network oscillators, and sensory feedback interact to enable rhythmic motor generation and coordination? The identification and study of locomotor Central Pattern Generators (CPGs) in the mammalian spinal cord has been complicated by the system's complexity and the large numbers of neurons that are potentially involved. As a result, many components of the mammalian locomotor rhythm generator remain unidentified (*Kiehn, 2006*; *Mullins et al., 2011*; *Kiehn, 2016*). However, work on vertebrate and invertebrate models, such as swimming leeches and lampreys, has allowed the basic principles and components of neural oscillators to be identified (*Goulding, 2009*; *Mullins et al., 2011*).

Electrophysiological studies on the leech isolated ventral nerve cord (VNC) have found that individual ganglia distributed along the body can generate oscillatory patterns that mimic those of normal swimming (*Weeks, 1981*; *Kristan et al., 2005*; *Marder et al., 2005*). Stretch sensation and central control couple the oscillatory units in both the ascending and descending directions, such that the intact animal's entire circuit functions in synchrony during swimming (*Mullins et al., 2011*).

In the lamprey, excitatory interneurons proposed to be rhythm generators are also found throughout the approximately 100 spinal segments, which can generate oscillations when isolated (*Mullins et al., 2011*; *Kiehn, 2016*). The distributed nature of rhythm generation in swimming models bears some resemblance to that found in hindlimb locomotion in limbed vertebrates, for which rhythm generating capability is distributed along the caudal spinal cord (*Kiehn, 2006*). Moreover, analogues of many of the key neuronal classes underlying these behaviors in lampreys and zebrafish are also found in the mouse spinal cord (*Goulding, 2009*; *Kiehn, 2016*).

Despite these findings, a clear understanding of how motor systems generate locomotory oscillations at the network, cellular, and molecular levels remains elusive. In both the leech and lamprey, identification of the neurons responsible for rhythm generation remains incomplete, and the mechanism(s) by which these neurons generate swim rhythms are unclear (*Kristan et al., 2005*; *Mullins et al., 2011*). In the lamprey, proposed oscillator neurons have not been directly shown to generate the swimming rhythm (*Kiehn, 2016*). Moreover, the paucity of genetic manipulations available in these organisms makes it difficult to describe molecular mechanisms that contribute to rhythm generation.

The roundworm *C. elegans* is a promising model for achieving an integrated behavioral, circuit, and molecular understanding of how locomotion is generated and coordinated. *C. elegans* has a compact nervous system containing a few hundred neurons, for which a nearly complete wiring diagram of synaptic connectivity has been mapped (*White et al., 1986*; *Varshney et al., 2011*). Worms' optical transparency allows researchers to monitor neural activity with genetically encoded calcium and voltage sensors (*Kerr et al., 2000*; *Kerr, 2006*; *Flytzanis et al., 2014*), and manipulate neurons and muscles using optogenetics (*Nagel et al., 2005*; *Zhang et al., 2007*; *Leifer et al., 2011*; *Stirman et al., 2011*; *Husson et al., 2012*; *Kocabas et al., 2012*; *Fang-Yen et al., 2015*; *Gao et al., 2015*). *C. elegans* is readily amenable to a powerful set of genetic manipulations (*Ahringer, 2006*; *Evans, 2006*) and shares extensive genetic homology with humans (*Lai et al., 2000*). Classical neurotransmitters involved in *C. elegans* locomotion include acetylcholine (*Rand, 2007*), GABA (*Jorgensen, 2005*), glutamate (*Brockie and Maricq, 2006*), and the biogenic amines dopamine and serotonin (*Chase and Koelle, 2007*).

*C. elegans* moves forward by generating sinusoidal dorso-ventral bending waves that propagate from anterior to posterior. The circuit for locomotion consists of interneurons, excitatory and inhibitory motor neurons, and body wall muscles (*White et al., 1976*; *Chalfie et al., 1985*; *White et al., 1986*; *Altun and Hall, 2011*). The majority of motor neuron cell bodies are located in the ventral nerve cord (VNC), which runs along the ventral side of the body from head to tail (*White et al., 1986*; *Altun and Hall, 2011*). The VNC motor neurons include A, B, VC, D, and AS cell types. Laser ablation studies have shown that the A-type neurons are essential for reverse locomotion, whereas the B-type are required for forward locomotion (*Chalfie et al., 1985*). The D-type (GABAergic) motor neurons are required for a normal amplitude of body bending waves but are not essential for locomotion itself (*McIntire et al., 1993b*). The function of the AS neurons is unknown. The VC neurons are involved in egg laying (*Waggoner et al., 1998*). These classes all form neuromuscular junctions with body wall muscles (BWMs).

While the basic architecture of the motor circuitry has been delineated by laser ablation studies, much less is understood about how its components interact to generate coordinated locomotory behavior. Perhaps most notably, it is not known which elements generate the worm's dorso-ventral oscillations during forward movement, nor how many such rhythm generators exist. Worms are capable of limited movement despite ablation of most premotor interneurons (*Chalfie et al., 1985*; *Wicks and Rankin, 1995*; *Zheng et al., 1999*). When all premotor interneurons are removed, animals did not generate directional movement, but retained the ability to generate local body bends (*Kawano et al., 2011*). However, forward locomotion was observed after ablation of all premotor interneurons and A motor neurons (Gao et al, 2017), suggesting that periodic bending during forward locomotion may be organized at the level of the non-A motor neurons and/or the body wall muscles.

Sensory feedback has been shown to play an important role in coordinating *C. elegans* motor behavior. The frequency of *C. elegans* undulation depends continuously on mechanical loading by its environment (*Berri et al., 2009*; *Fang-Yen et al., 2010*), and computational models based on proprioceptive feedback and coupling have recapitulated key aspects of locomotory behavior (*Boyle et al., 2012*; *Wen et al., 2012*). Experiments in which the worm's body was partially immobilized in a microfluidic device showed that the posterior B-type motor neurons mediate anterior-to-posterior proprioceptive coupling (*Wen et al., 2012*). B-type motor neurons sense the body curvature and induce bending in the same direction (ventral or dorsal) posterior to the sensed bending.

These findings suggested a model for forward locomotion, similar to one proposed earlier (*Karbowski et al., 2008*), in which a single rhythm generator generates bending undulations in the head, and these undulations propagate through the body from anterior to posterior via proprioceptive coupling (*Wen et al., 2012*). This model successfully reproduced the continuous variation in locomotory characteristics observed in varied mechanical environments (*Berri et al., 2009*; *Fang-Yen et al., 2010*). This work, while demonstrating how a wave can be propagated along the body, did not directly address the identity of the rhythmic generator(s). Furthermore, it focused on coupling in the posterior of the worm and did not determine whether head and neck proprioception is similarly essential for bending wave propagation.

How might the locomotory circuit be organized? The circuit contains one or more oscillators (*Figure 1*). A model including a single oscillator with proprioceptive coupling (*Figure 1C*) predicts that a disruption in body bending at any location will inhibit posterior bending (*Figure 1E*). Alternative possibilities (*Gjorgjieva et al., 2014*; *Zhen and Samuel, 2015*) include the presence of multiple oscillators distributed along the motor circuit (*Figure 1D*), as in the vertebrate spinal cord (*Kiehn, 2006*; *Mullins et al., 2011*; *Kiehn, 2016*) and in the VNC of some invertebrates (*Kristan et al., 2005*). These oscillatory units could be capable of generating undulations in the posterior of the worm even if anterior neural activity or physical bending is interrupted (*Figure 1F*).

In this work, we used spatiotemporally targeted optogenetic illumination (*Leifer et al., 2011*; *Stirman et al., 2011*) and lesion studies to show that the mid-body VNC motor circuit contains multiple units capable of independent oscillation. We found a fundamental architecture in the *C. elegans* motor circuit similar to that previously described in other vertebrate and invertebrate models.

## Results

### Rhythmic posterior undulation persists despite anterior paralysis

We first sought to test a model in which there is a single oscillator in the head and proprioceptive feedback is the dominant organizer of bending waves along most of the body (*Figure 1C*). This model, supported by experiments showing that immobilization of the mid-body of worms induced the posterior to adopt the same direction of curvature as the immobilized region (*Wen et al., 2012*), predicts that paralysis of any region will eliminate undulations posterior to the paralyzed region (*Figure 1E*). In particular, we asked whether paralysis of the head and 'neck' (a region immediately posterior to the head) would halt body bending posterior to these regions.

To manipulate neural and muscular activity in freely moving worms, we constructed an optogenetic targeting system similar to that previously described (*Leifer et al., 2011*). Briefly, this system uses real-time imaging processing and a digital micromirror device to project laser illumination onto arbitrarily specified regions of an unrestrained worm.

To examine the effect of inhibiting anterior muscles, we first used this system to project 532 nm illumination onto worms expressing the inhibitory opsin halorhodopsin (NpHR/Halo) in all body wall muscles under the control of the *myo-3* promoter (*Zhang et al., 2007*; *Leifer et al., 2011*). We quantified the movement of worms before and during optogenetic manipulation by measuring the curvature of the worm over time (*Figure 1A,B*) (*Pierce-Shimomura et al., 2008*; *Fang-Yen et al., 2010*; *Leifer et al., 2011*; *Wen et al., 2012*). We specify longitudinal positions via a body coordinate ranging from 0 at the tip of the head to 100 at the end of the tail.

Illuminating body coordinates 50–65 in P*myo-3::NpHR* worms caused substantial paralysis in the tail (not shown), consistent with previous findings (*Leifer et al., 2011*; *Wen et al., 2012*). When we paralyzed the anterior 33% or 45% of the worm, however, we observed robust oscillations in

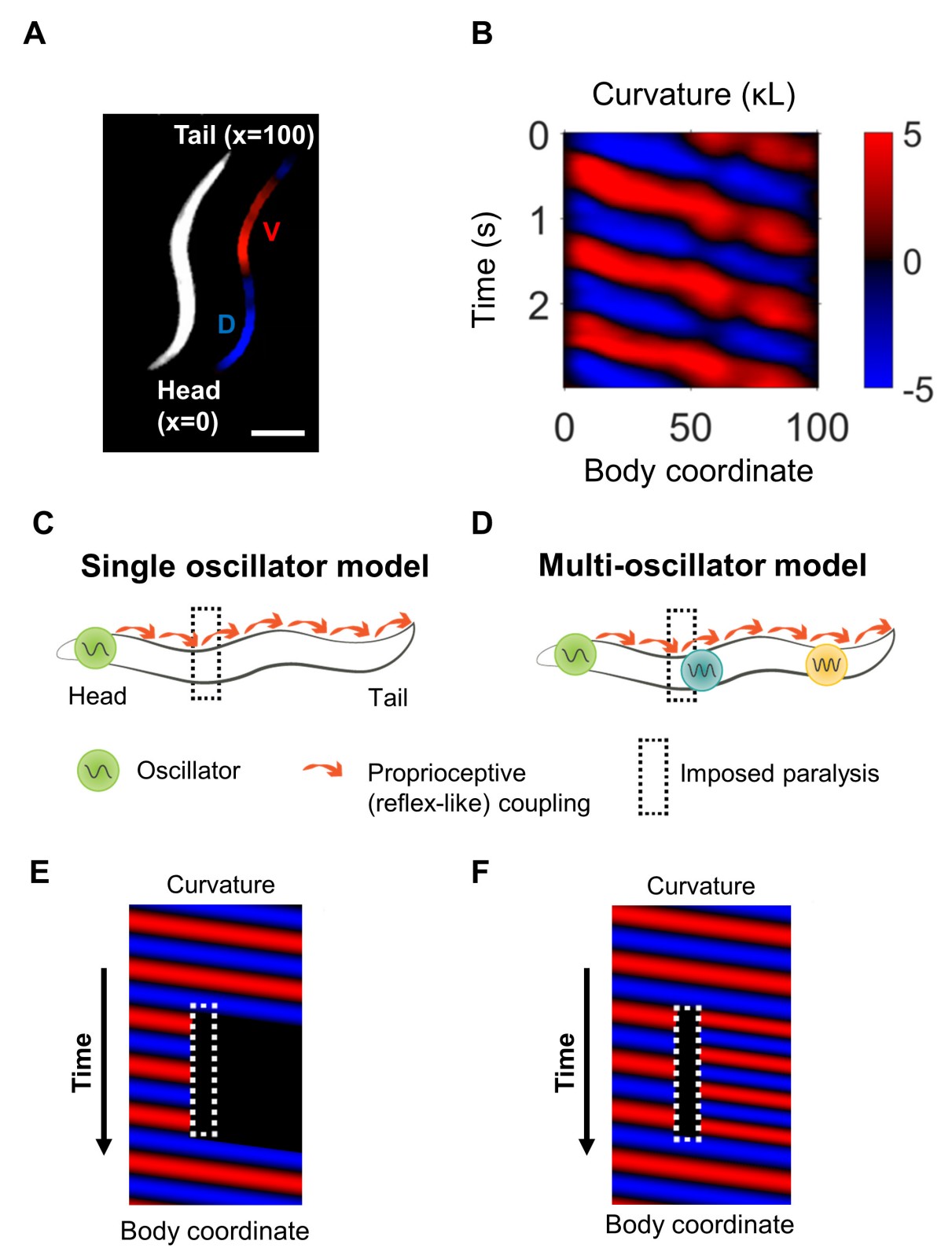

**Figure 1.** Overview of curvature analysis and models of rhythm generation. (A) Dark field image of a worm shown with curvature segmentation. Dorsal bending is shown in blue and ventral bending in red. The dorso-ventral orientation is arbitrary unless otherwise specified. The worm's centerline is used to define a coordinate system in which the head and tail are located at body coordinates 0 and 100, respectively. Scale bar: 200 μm. (B) Curvature map from a normally swimming worm. The curvature at time t = 0 s corresponds to the image shown in (A). (C) In a single-oscillator model of locomotion, an

*Figure 1 continued on next page*

*Figure 1 continued*

unknown oscillator causes rhythmic head bending, and a reflex-like coupling mechanism mediates propagation of these bends along the rest of the body. (D) A multi-oscillator model (*Gjorgjieva et al., 2014*) posits the existence of additional circuit units outside the head capable of generating oscillations. (E) Conceptual curvature map showing predicted worm behavior after paralyzing a small region of the body (dotted white box). The single-oscillator model predicts that all regions posterior to the paralyzed region will also become paralyzed. (F) Conceptual curvature map predicting the outcome of the same manipulation applied to a multi-oscillator model. If additional oscillators exist posterior to the paralyzed region, additional tail oscillations may arise, potentially with different amplitude, frequency, and/or phase.

DOI: https://doi.org/10.7554/eLife.29913.002

posterior regions of the body. In addition, we found to our surprise that illumination of the anterior 33% of the body caused the tail's undulation frequency to increase (*Figure 2A,D*; *Video 1*).

Next, we asked whether oscillations in the posterior would persist under optogenetic inhibition of excitatory motor neurons instead of inhibition of muscles. We illuminated worms expressing NpHR in all cholinergic neurons (P*unc-17::NpHR*), including the A-type and B-type motor neurons, head motor neurons, and several other neuronal cell types (*Duerr et al., 2008*). We found that while optogenetic inhibition of cholinergic neurons in the head and neck caused anterior paralysis, tail undulation often persisted (*Figure 2B,D*, *Video 1*).

During optogenetic muscle or neuron inactivation, the amplitude of the bending wave in the head decreased greatly but did not vanish (*Figure 2A,B,E*), leaving open the possibility that a residual small amplitude wave allows propagation of the bending wave through the partially paralyzed region. We therefore sought means of paralyzing the head more effectively.

We hypothesized that regional paralysis could be induced by lesioning the anterior BWMs instead of hyperpolarizing them. To selectively lesion muscles, we used region-targeted illumination at 470 nm of P*myo-3::PH::miniSOG* worms in which the photosensitizing protein miniSOG is expressed in body wall muscles (*Xu and Chisholm, 2016*). The anterior portion of most treated animals was nearly immobile (*Figure 2C*, *Video 1*, especially the last 8 s). Nevertheless, undulation posterior to the region of illumination was routinely observed in these animals.

We also conducted thermal lesioning experiments in which touched the anterior half of the worm with a hot platinum wire attached to a soldering iron. After this treatment, the animal's head and neck were again nearly motionless, yet rhythmic undulation routinely persisted in the tail (*Figure 2—figure supplement 1*, *Video 1*).

Our finding that posterior undulation can persist despite anterior paralysis is consistent with a multi-oscillator model (*Figure 1D*) and not with a single oscillator model that relies on reflex-like signaling for wave propagation (*Figure 1C*).

## The head and tail are capable of simultaneous oscillations at different frequencies

The finding that optogenetic inhibition of anterior muscles induces higher frequency oscillations in the tail suggests that an interruption of propagating activity in the motor circuit enables independent activity in a posterior oscillator. To test this idea further, we applied several optogenetic manipulations to inhibit motor coupling in the neck only, leaving the head and tail free to oscillate.

First, we optogenetically inhibited neck muscles in P*myo-3::NpHR* worms. In most trials, optogenetically inhibiting neck muscles prevented waves generated in the head from propagating through the neck. During the interruption of these waves, the tail exhibited bending undulations at a higher frequency than that of the head, resulting in the animal simultaneously undulating at two distinct frequencies (*Figure 3A*, *Video 2*). We henceforth refer to this behavior, whether or not induced by any manipulation, as two-frequency undulation (2FU).

We observed 2FU upon inhibiting all neck cholinergic neurons (*Figure 3B*, *Video 2*) and also upon inhibiting neck B-type motor neurons (P*acr-5::Arch*; *Figure 3C*, *Video 2*). These manipulations led to a large decrease in wave amplitude in the neck and a smaller decrease in wave amplitude in the tail (*Figure 3F*). Nevertheless, multiple animals in each experiment showed 2FU, with the highest ratios of tail frequency to head frequency seen in worms in which the neck muscles were inhibited (*Figure 3E*).

The bending amplitude of the tail generally decreased as the frequency increased (*Figure 3E,F*), consistent with the changes in bending frequency and amplitude previously observed when the

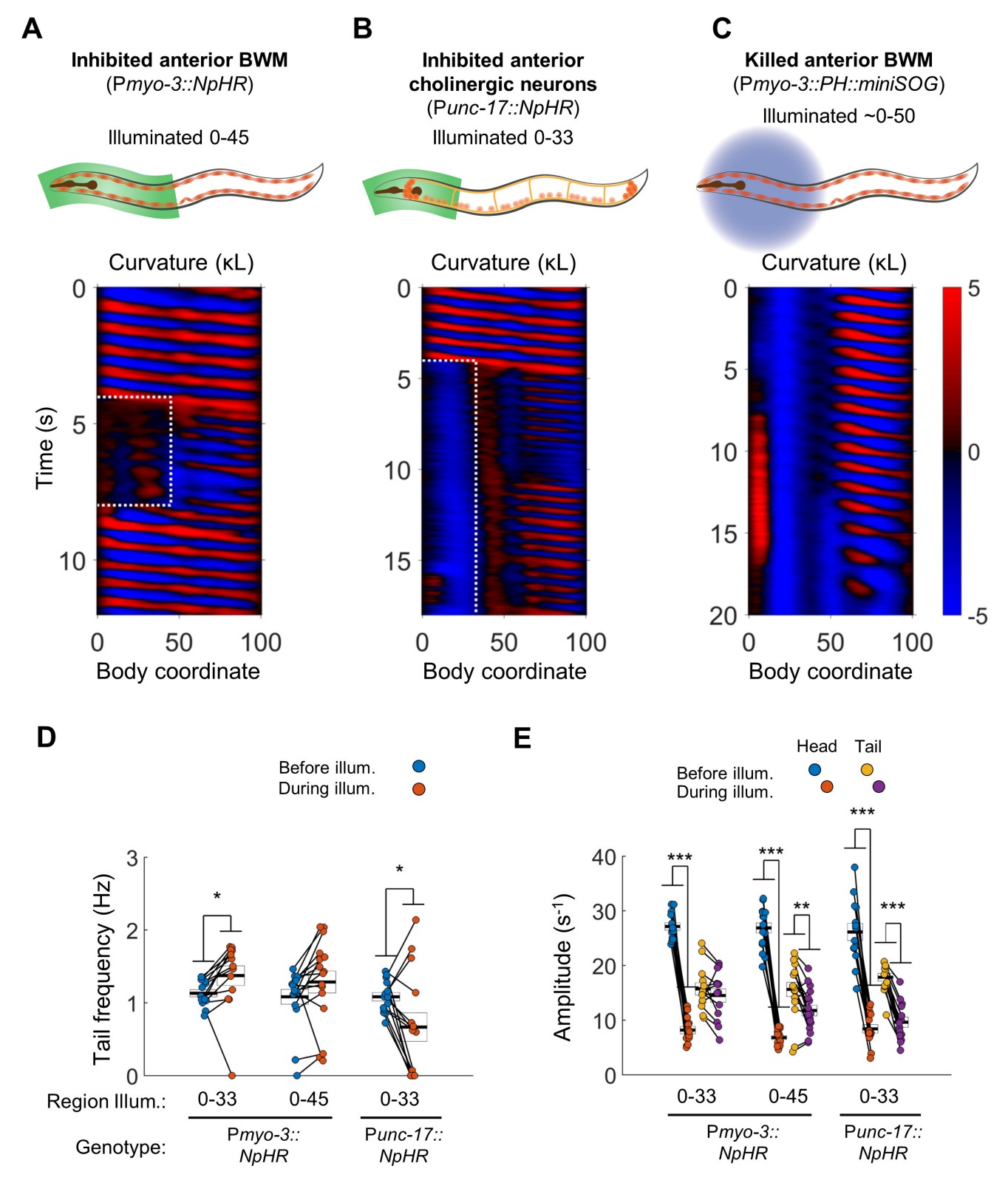

**Figure 2.** Anterior undulation is not required for posterior undulation. (**A**) Inhibition of anterior BWMs (via P*myo-3::NpHR*) increases tail frequency. Body coordinates 0–45 were illuminated with green light (532 nm wavelength) to trigger relaxation of the anterior muscles. The spatiotemporal extent of green laser illumination is indicated by the white dotted box. (**B**) Inhibition of anterior cholinergic neurons (via P*unc-17::NpHR*; P*unc-17::ChR2*) does not prevent tail undulation. Body coordinates 0–33 were illuminated with green light to optogenetically inhibit anterior motor activity. (**C**) Tail undulations

*Figure 2 continued on next page*

**Figure 2 continued**

persist despite paralysis of the anterior BWMs due to miniSOG-mediated lesion of muscle cells. Animals were subjected to mechanical stimulation to induce locomotion (see Materials and methods). A total of nine animals were illuminated with blue light (472 nm wavelength) in approximately their anterior halves. Of these, five displayed partial-body forward swimming as depicted here, three were immobile, and one was not sufficiently paralyzed in the head. Six control worms, which were mounted identically but not illuminated, all displayed waves propagating normally from head to tail (not shown). (D) Inhibition of some anterior muscles (body coordinate 0–33, N = 10 worms) significantly increases tail frequency. Inhibition of most anterior muscles (0–45, N = 10 worms), or inhibition of anterior cholinergic neurons (N = 14 worms) produces mixed results; some animals generate high frequency tail oscillations while others slow down. Each colored circle represents one trial; worms may have multiple trials. Tail frequency is measured at body coordinate 85. Error boxes represent the mean and SEM. (E) Amplitude of undulation in the head and tail before and during muscle or neuron inhibition. Head frequency is measured at body coordinate 15. Note sharp decreases in head amplitude during all three manipulations. Amplitude here and henceforth is measured as the root mean square of the time derivative of the curvature times worm length $\mathrm{rms}\left(L \cdot \frac{d\kappa}{dt}\right)$ and has units of $s^{-1}$. (*) p<0.05; (**) p<0.01; (***) p<0.001; paired t-test.

DOI: https://doi.org/10.7554/eLife.29913.003

The following figure supplement is available for figure 2:

**Figure supplement 1.** Tail undulation after gross head lesioning.

DOI: https://doi.org/10.7554/eLife.29913.004

viscosity of a fluid environment was varied (*Fang-Yen et al., 2010*). The opposite trends of amplitude and frequency may reflect a constraint to the maximum absolute rate of change of curvature, which is proportional to the product of amplitude and frequency.

In some experiments, the optogenetic manipulation of motor neurons or muscles did not completely block wave transmission through the paralyzed region. In these trials, some tail waves appeared synchronized with head waves, whereas others did not (*Figure 3C*). To test whether 2FU can occur after stronger disruption of motor coupling, we lesioned mid-body muscles in P*myo-3::PH::miniSOG* worms. This manipulation indeed led to stronger decoupling between head and tail oscillations, but did still not prevent 2FU (*Figure 3D*, *Video 2*).

If the posterior motor circuit of B and AS type neurons contains additional oscillating units, we reasoned that localized undulations might occur after selectively activating small portions of the motor circuit while inhibiting the rest. We therefore examined worms in which both the inhibitory opsin NpHR and the excitatory opsin Channelrhodopsin-2 (ChR2) were expressed in the cholinergic neurons, after the A-type motor neurons were ablated by P*unc-4::miniSOG*.

We first illuminated these animals with 590 nm wavelength (yellow) light throughout the body to inhibit all cholinergic neurons. While maintaining this yellow illumination, we targeted small portions of the tail with 473 nm wavelength (blue) light, activating ChR2 and stimulating a few posterior B and AS neurons. Under these conditions, several animals generated high-frequency localized undulations in the tail (*Figure 3—figure supplement 1A*, *Video 3*). These findings further support the presence of additional oscillator(s) in this region.

If multiple independent oscillators underlie a worm's forward movement under physiological conditions, we reasoned that independent head and tail oscillations might also be observable in animals without induced lesions or optogenetic perturbations.

Wave frequency depends strongly on the degree of mechanical loading from the environment, for example decreasing with viscosity of the fluid medium (*Berri et al., 2009*; *Fang-Yen et al., 2010*). We hypothesized that head and tail oscillations might be decoupled by placing the anterior and posterior of a worm in fluids of different viscosities. When we studied worms transitioning between regions of a low-viscosity buffer into highly viscoelastic hydroxypropylmethylcellulose (HPMC) islands (see Methods), we observed 2FU in 6 of 41 worms (15%). In these

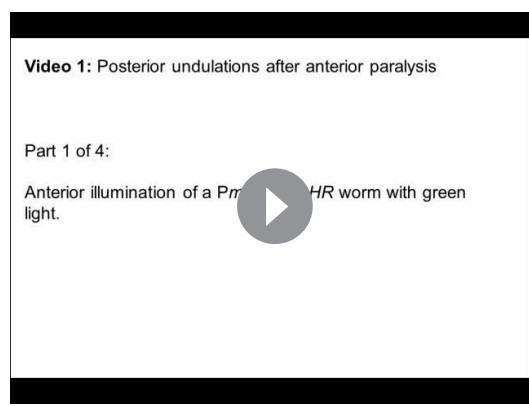

**Video 1.** Posterior undulations after optogenetic inhibition of anterior body wall muscles or cholinergic neurons.

DOI: https://doi.org/10.7554/eLife.29913.005

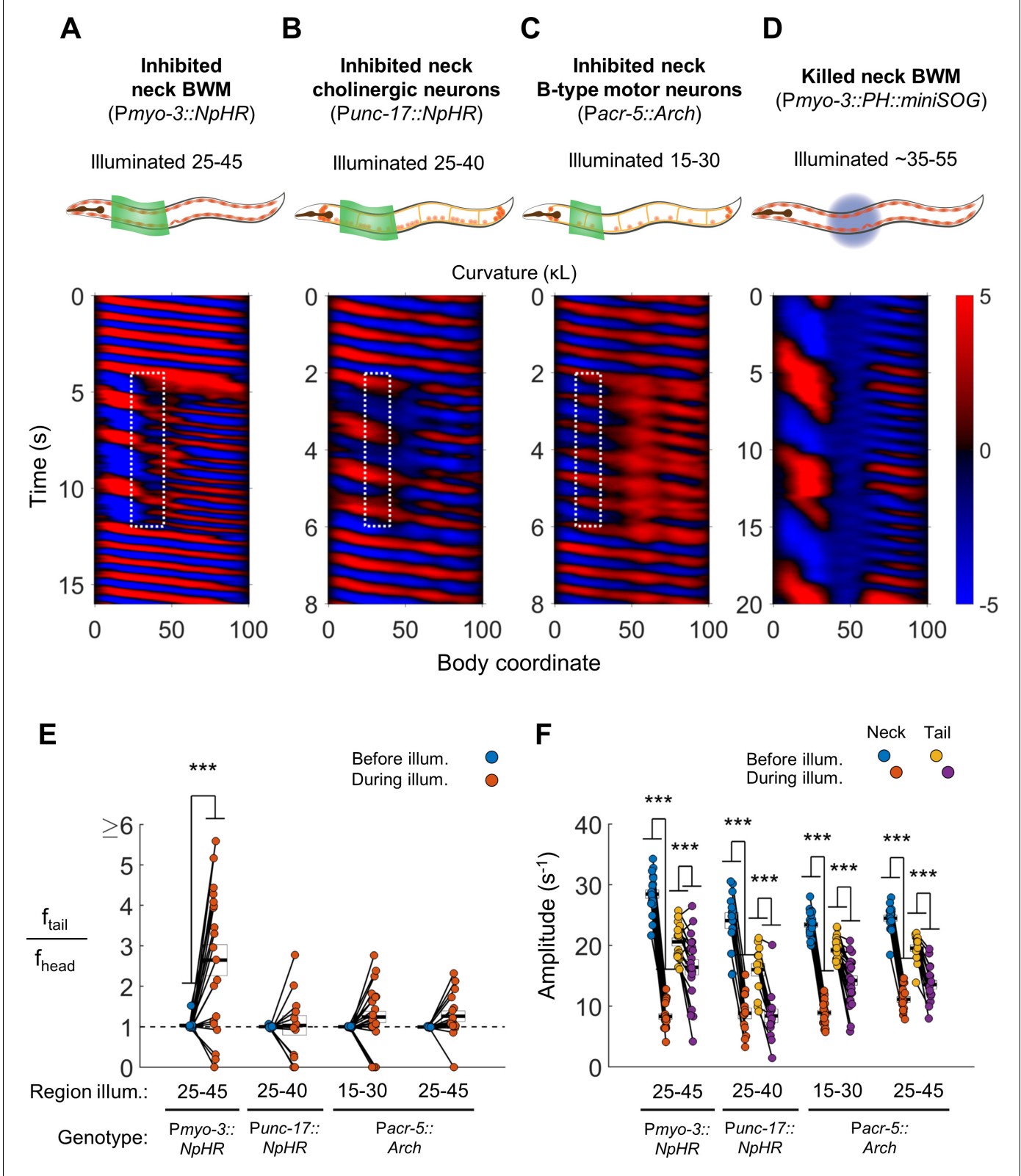

**Figure 3.** Disruption of motor coupling in the neck de-synchronizes head and tail oscillations. (**A**) Inhibition of neck BWMs (via P*myo-3::NpHR*) increases tail frequency and decreases head frequency. We refer to this effect as two frequency undulation (2FU). Body coordinates 25–45 were illuminated with green light to induce relaxation of neck muscles. The spatiotemporal extent of green laser illumination is indicated by the white dotted box. (**B,C**) Inhibition of neck cholinergic neurons (P*unc-17::NpHR*) or neck B-type motor neurons (P*acr-5::Arch*) also induces 2FU behavior. (**D**) Two frequency
*Figure 3 continued on next page*

*Figure 3 continued*

undulation after miniSOG-induced paralysis of the mid-body BWMs. Animal was subjected to mechanical stimulation to induce locomotion, but also displayed this behavior prior to stimulation. A total of 10 individuals were illuminated with blue light on approximately one-fifth of their body length, centered near the vulva. Of these, seven displayed 2FU as depicted here, one was immobile, and two were not sufficiently paralyzed in the mid-body to disrupt bending waves. Color map data are scaled down by 50% because bends in this animal had higher amplitudes than those shown in A-C. (E) Several optogenetic manipulations produced decoupled head and tail oscillation. 2FU is assayed by dividing tail frequency by head frequency in each worm. Before illumination, the head (body coordinate 15) and tail (body coordinate 85) usually oscillate at the same frequency. During illumination, tail frequency often exceeds head frequency. Each colored circle pair represents one trial; worms may have multiple trials. N = 11, 10, 12, and 10 worms per condition, respectively. Error boxes represent the mean and SEM. (F) Amplitude of undulation in the neck and tail before and during neck muscle or neuron inhibition. Neck amplitude is measured at body coordinate 35. (*) p<0.05; (**) p<0.01; (***) p<0.001; paired t-test.

DOI: https://doi.org/10.7554/eLife.29913.006

The following figure supplement is available for figure 3:

**Figure supplement 1.** Additional disruptions to motor coupling cause 2FU.

DOI: https://doi.org/10.7554/eLife.29913.007

---

animals, the tail continued oscillating at a high frequency for at least two full cycles even as the head frequency was sharply reduced (*Figure 3—figure supplement 1B*, *Video 3*). Although these events were uncommon, they demonstrated that 2FU can occur in *C. elegans* with no internal perturbations.

Taken together, these results strongly suggest that the *C. elegans* forward motor circuit contains at least two units capable of independent rhythm generation and that a partial breakdown in anterior proprioceptive coupling (for example by inhibiting neck BWMs) is sufficient to reveal the presence of the posterior oscillating unit(s).

## Most premotor interneurons are not essential for rudimentary forward movement or 2FU

To better understand the source of tail oscillations during 2FU, we used genetic analysis and lesion studies to ask which components of the motor circuit are required for this behavior. Almost all chemical or electrical synaptic connections to the VNC motor neurons are made by the premotor interneuron (IN) classes AVB, PVC, AVA, AVD, and AVE (*White et al., 1986*). Laser ablation studies have indicated that AVB, and to a lesser degree PVC, are essential for normal forward locomotion (*Chalfie et al., 1985*), although rudimentary forward crawling is possible in their absence if the reverse-driving A motor neurons are also removed (*Gao et al., 2017*). Therefore, we asked whether 2FU is possible in the absence of AVB, PVC, and all other premotor INs.

To determine if the premotor interneurons are required for 2FU, we first asked whether optogenetic muscle inhibition in the neck in worms lacking premotor interneurons would induce 2FU (c. f. *Figure 3A,E*). We used transgenic strains in which expression of the apoptosis-promoting interleukin-converting enzyme (ICE) was used to ablate premotor INs and some other neurons (*Zheng et al., 1999*). When ICE is expressed under the control of the *nmr-1* or *glr-1* promoters, the PVC, AVA, AVD, and AVE interneurons are removed. AVB, however, are present in both P*nmr-1*::ICE (*Kawano et al., 2011*), and P*glr-1*::ICE worms (Kawano, Po, and Zhen, personal communication). We generated the strains P*myo-3*::NpHR; P*glr-1*::ICE and P*myo-3*::NpHR; P*nmr-1*::ICE. We found that both strains were capable of 2FU during optogenetic inhibition of neck muscles (*Figure 4A,E*, *Figure 4—figure supplement 1A,E*). This result demonstrates that 2FU does not require most premotor interneurons, including the forward locomotory interneurons neurons PVC.

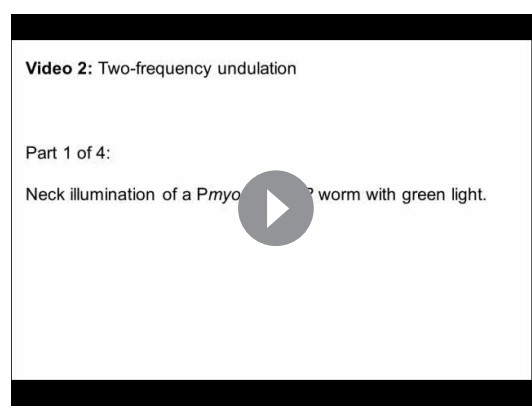

**Video 2.** Two-frequency undulation during optogenetic inhibition of neck BWM, cholinergic neurons, or B motor neurons.

DOI: https://doi.org/10.7554/eLife.29913.008

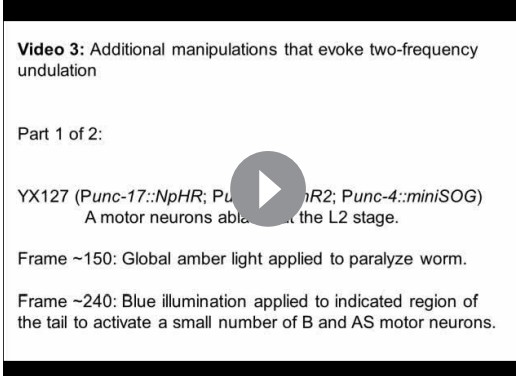

**Video 3.** Additional manipulations that evoke two-frequency undulation (2FU): Stimulation of B and AS during tail paralysis; inhomogenous mechanical environment.
DOI: https://doi.org/10.7554/eLife.29913.009

The interneuron AVB is coupled to the B-type motor neurons by an extensive network of gap junctions. Formation of these connections, as well as electrical coupling between the premotor interneurons and the motor neurons of the circuit for reverse locomotion, requires UNC-7 expression in AVB and UNC-9 expression in the B-type motor neurons (*Starich et al., 2009*; *Kawano et al., 2011*; *Liu et al., 2017*). UNC-9 also participates in electrical coupling between BWM cells (*Liu et al., 2006*). We asked whether animals lacking UNC-7 or UNC-9 could exhibit 2FU. We crossed *unc-7* and *unc-9* mutations into our P*myo-3::NpHR* strain and performed optogenetic experiments as before. The *unc-7* and *unc-9* worms are uncoordinated and exhibit significantly reduced spontaneous forward locomotion (*Starich et al., 2009*; *Kawano et al., 2011*). To initiate a short bout of forward locomotion, we used a cell phone motor to apply a mechanical stimulus in the form of a 3–5 s, ≈200 Hz vibration of the slide just before illumination. We found that both strains were capable of 2FU, although it appeared to occur less often than in PVC-ablated animals, and sometimes occurred prior to neck muscle inhibition (*Figure 4B,E*; *Figure 4—figure supplement 1E*; *Figure 4—figure supplement 5A, B*). This finding shows that the UNC-7/UNC-9 gap junctions, including those between AVB and the B-type motor neurons, are not required for 2FU.

Finally, we ablated AVB, labeled by P*sra-11::D3cpv* (*Kawano et al., 2011*), using a pulsed infrared laser ablation system (*Churgin et al., 2013*) that we modified to intentionally lesion tissue (see Materials and methods). This procedure generally removed both AVB cell bodies and their associated processes, and possibly other head neurons, but not PVC. When subjected to the same experiment as described above, 2FU events were nearly diminished. Very rarely we observed uncoupled undulation events that were not correlated with neck muscle inhibition (*Figure 4—figure supplements 1B, E*, *2* and *5C*). Taken together, these results suggest that most individual classes of premotor interneurons, including AVA, AVD, AVE, and PVC, are not essential for 2FU. However, AVB may play a key role in activating the rhythm generator(s) to allow oscillation.

## Several classes of motor neurons are not required for forward locomotion or 2FU

The premotor interneurons comprise the primary circuit connection between the VNC motor neurons and the worm's other sensory and interneuronal circuits (*White et al., 1986*). The finding that most premotor interneurons are not necessary for forward locomotion and 2FU suggests that high-frequency tail undulations during 2FU may originate from the motor neurons themselves. We asked, in the presence of all premotor interneurons, whether any classes of motor neurons are required for 2FU.

We first examined the A-type motor neurons. While the A class motor neurons are preferentially active during reverse locomotion (*Haspel et al., 2010*; *Kawano et al., 2011*) and are required for reverse locomotion (*Chalfie et al., 1985*), it is conceivable that they play a role in 2FU. We found that ablating the A- and VC-type motor neurons with a genetically targeted ROS generator (P*unc-4::miniSOG*) did not prevent 2FU induced by neck paralysis during forward locomotion (*Figure 4—figure supplement 1C,E*). This result supports the idea that the A and VC motor neurons are not necessary for the posterior forward oscillator(s).

Next, we examined whether the GABAergic D-type motor neurons are required for 2FU. The D-type motor neurons release GABA onto the UNC-49 receptor to trigger contralateral muscle inhibition during a bend. Therefore, the putative null allele *unc-49(e407)* (*Bamber et al., 1999*; *Liewald et al., 2008*) should effectively block the functional output of D-type motor neurons. Indeed, *unc-49* mutants exhibited simultaneous dorsal and ventral contractions when stimulated for

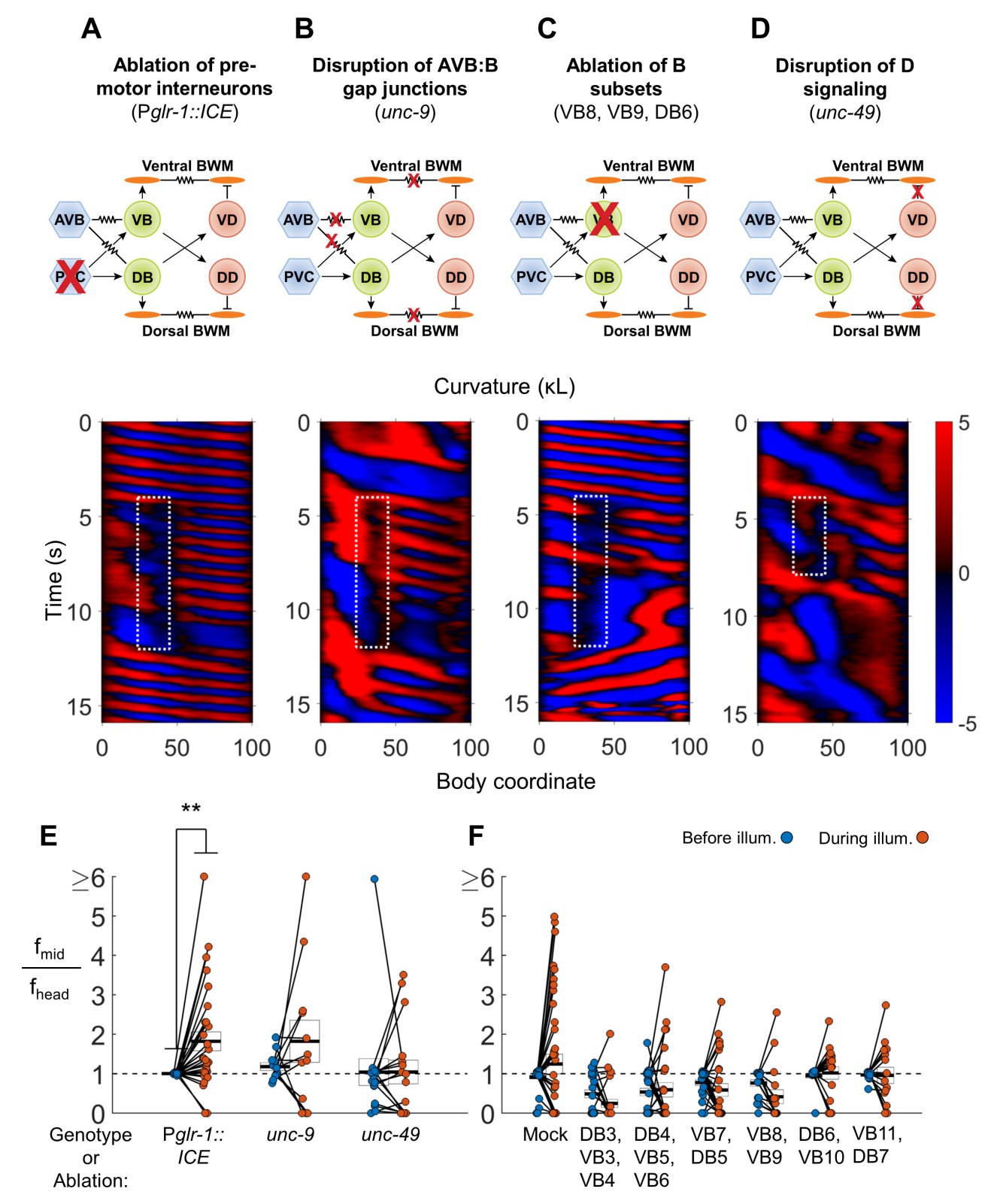

**Figure 4.** VNC premotor interneurons, D-type motor neuron signaling, and individual B-type motor neurons are not required for 2FU. (**A**) 2FU occurs despite ablation of premotor interneurons (P*glr-1::ICE*; P*myo-3::NpHR*). The spatiotemporal extent of green laser illumination is indicated by the white dotted box. (**B**) 2FU occurs despite disruption of AVB:B gap junctions and BWM:BWM gap junctions (*unc-9*; P*myo-3::NpHR*). (**C**) 2FU occurs despite ablation of a small subset of the B-type motor neurons (P*myo-3::NpHR*; P*acr-2::wCherry*). VB8, VB9, and DB6 were ablated using our pulsed infrared

*Figure 4 continued on next page*

*Figure 4 continued*

laser system. (D) 2FU occurs despite elimination of GABAergic signaling (*unc-49*; P*myo-3::NpHR*). (E) When subjected to neck paralysis (P*myo-3::NpHR*; P*acr-2::wCherry*), 2FU occurs reliably in P*glr-1::ICE* animals and occasionally in *unc-9* and *unc-49* animals. Each colored circle pair represents one trial; worms may have multiple trials. N = 12, 8, and 10 worms per condition, respectively. Head frequencies are measured at body coordinate 15. Mid-body are frequencies are measured at body coordinate 60. Error boxes represent the mean and SEM. (*) p<0.05; (**) p<0.01; (***) p<0.001; paired t-test. (F) When subjected to neck paralysis (P*myo-3::NpHR*), 2FU occurs at least occasionally despite ablation of subsets of the B-type motor neurons by our pulsed infrared laser system. For each condition, data are only considered from worms that have all specified neurons missing; some worms in each group may have additional B-type or other neurons missing. N = 40, 30, 32, 27, 18, 18, and 16 trials from 10, 10, 10, 8, 7, 7, and 7 worms per condition, respectively. Mean mid-body/head frequency ratios during illumination are significantly lower than mock controls for all ablation conditions except DB6, VB10 and VB11, DB7 (p<0.05 by one-way ANOVA with Bonferroni post-hoc comparisons).
DOI: https://doi.org/10.7554/eLife.29913.010

The following figure supplements are available for figure 4:

**Figure supplement 1.** VNC premotor interneurons and several VNC motor neuron classes are not required for de-synchronized oscillations.
DOI: https://doi.org/10.7554/eLife.29913.011

**Figure supplement 2.** Ablation of the AVB interneurons.
DOI: https://doi.org/10.7554/eLife.29913.012

**Figure supplement 3.** Subsets of B-type motor neurons are not required for 2FU.
DOI: https://doi.org/10.7554/eLife.29913.013

**Figure supplement 4.** B-type motor neurons posterior to the vulva are not required for 2FU.
DOI: https://doi.org/10.7554/eLife.29913.014

**Figure supplement 5.** Examples of 2FU occurrence prior to optogenetic inhibition of neck muscles.
DOI: https://doi.org/10.7554/eLife.29913.015

forward and reversal movement, as did animals in which D motor neurons were ablated (*McIntire et al., 1993a*). We found that animals harboring an *unc-49(e407)* mutation, while very slow swimmers, were nonetheless capable of 2FU during neck muscle paralysis (*Figure 4D,E*). In one case, we also observed 2FU before inhibition of neck muscles (*Figure 4—figure supplement 5D*).

The B-type motor neurons are required for forward locomotion (*Chalfie et al., 1985*; *Wen et al., 2012*), and are rhythmically active during forward locomotion (*Haspel et al., 2010*; *Kawano et al., 2011*; *Wen et al., 2012*). We sought to determine whether any individual or small group of these neurons is essential for 2FU. We ablated groups of 2–6 B-type motor neurons at a time using our pulsed infrared laser system. 2FU was observed very rarely after ablating DB3, VB3, and VB4 or VB8 and VB9 (*Figure 4C,F*), although in both conditions there were additional instances of 2FU that occurred outside the time 3 s window used for frequency analysis (*Fouad et al., 2017*). 2FU occurred repeatedly, although not commonly, in all other conditions. (*Figure 4F*; *Figure 4—figure supplements 3* and *4*). In nearly all cases, ablation of a DB motor neuron resulted in the disappearance of its commissural process, but we could not determine whether ablated VB neuronal processes were similarly removed. Therefore, we cannot completely exclude the possibility that a specific neuronal process can generate rhythms for 2FU in the absence of its associated cell body.

Taken together, these results suggest that in the presence of premotor interneurons, the B class motor neurons contribute to neck-paralysis induced 2FU, but no single member is essential for generating high-frequency tail rhythms during 2FU. The A, VC and D motor neurons are not required for 2FU. The AS motor neurons were not investigated. Our results are consistent with a model in which posterior rhythm generation can arise from multiple subsets of B or AS motor neurons.

## B-type motor neurons, as a class, are essential for 2FU

We next asked whether the B motor neurons as a class are required for 2FU. We first considered *vab-7* mutants, in which the DB motor neurons have aberrantly reversed processes. These worms have disrupted wave propagation in the tail, which coils towards the ventral side (*Wen et al., 2012*). We found that *vab-7* mutants had mildly or strongly paralyzed tails and were incapable of 2FU when neck muscles were inhibited, suggesting that *vab-7* is essential for 2FU (*Figure 4—figure supplement 1D,E*).

The inability of *vab-7* worms to generate the high-frequency tail oscillation is consistent with the notion that broad disruption of the B motor neurons prevents 2FU. However, the behavioral deficit could also result from other effects of the mutation.

To ascertain whether the B motor neurons are required for 2FU, we performed broad ablations of the B motor neurons. We studied worms expressing P*unc-17β::PH::miniSOG* (**Xu and Chisholm, 2016**). In our integrated lines, we found that illumination of P*unc-17β::PH::miniSOG* worms preferentially eliminated the DA and DB over the VA and VB motor neurons (see Materials and methods). Worms in which most DB motor neurons were eliminated were dramatically less likely to show 2FU than mock controls, but were not incapable of doing so (**Figure 5**, **Video 4**; N = 9 out of 104 trials from 25 worms by blinded, randomized scoring). We performed the converse experiment – elimination of most VB motor neurons using our infrared laser system – and found the incidence of 2FU was again sharply reduced but not eliminated (**Figure 5**, **Video 4**, **Figure 5—figure supplement 1**). When we combined miniSOG and laser ablation to remove all DB and most VB motor neurons, animals were incapable of 2FU (**Figure 5**; 0 out of 102 trials from 27 worms by blinded, randomized scoring).

Taken together, these results suggest that the B motor neurons as a class are essential for independent tail undulation during forward movement. Our results are consistent with the hypothesis that the B motor neurons have a role in generating the high-frequency locomotory rhythm observed in 2FU.

## Multiple lesion-separated VNC segments are capable of independent rhythm generation

The observation that 2FU could persist despite disruptions to many components of the mid-body motor circuitry could also be explained by the hypothesis that additional rhythm generators located in the head are responsible for the observed high-frequency tail undulations. This possibility is supported by the findings that premotor INs account for the majority of synaptic inputs to the VNC motor neurons (**White et al., 1986**), and removal of the AVB premotor interneurons nearly abolished 2FU (**Figure 4—figure supplement 1E**).

To ascertain whether the mid-body motor circuit is capable of independent rhythm generation, we developed a method for eliminating synaptic connections between the mid-body motor neurons and the head circuits. We used our infrared laser system to sever both the VNC and the dorsal nerve cord (DNC) just posterior to the pharynx. In many cases, this procedure also severed other fasciculated process bundles that run parallel to the VNC and DNC (**Figure 6A**).

Several hours after disruption of the nerve cords, most animals were inactive (data not shown), but active forward locomotion was induced by application of a mechanical stimulus. Most of these worms could generate robust oscillations posterior to the cut location (**Figure 6A,E**, **Video 5**). Moreover, the tail often undulated at a higher frequency than the mid-body (**Figure 6E**). In these worms, oscillations in the head were highly disrupted. In some cases, low-amplitude waves propagating in the posterior-to-anterior direction occurred simultaneously with robust mid-body and tail waves propagating in the anterior-to-posterior direction (**Figure 6A**, **Video 5**), suggesting a deficit in coordination between circuits on either side of the lesion. These results suggest that synaptic connections from the head circuits to the motor neurons may not be essential for wave generation posterior to the head.

We considered the possibility that under these conditions, mid-body undulations were being caused by small movements in the head rather than generated by a second oscillator. To minimize the small movements of the head, we introduced an additional manipulation to reduce head movement. Using our infrared laser, we applied thermal damage to the worm's nerve ring (located in the head) in addition to cutting both nerve cords. Animals treated with these three lesions are henceforth referred to as 'VNC-lesioned' worms. These worms exhibited very little movement in the head. However, they continued to generate robust oscillations in the mid-body and even higher frequency oscillations in the tail (**Figure 6B,E**, **Video 5**). The pattern of locomotion in VNC-lesioned animals strongly resembled the 2FU induced by optogenetic perturbation, with the important difference that in our lesioned preparation, both frequencies were likely generated *outside* the head.

The emergence of multiple frequencies of undulation outside the head suggests that the VNC motor circuit itself may contain multiple units capable of independent oscillation. These units may exist in addition to any oscillating unit(s) in the head. To test this possibility directly, we cut the VNC and DNC in two locations: in the neck (anterior to VB3) and in the tail (posterior to VB8). We again thermally lesioned the head neurons to suppress head movement. Under these conditions, the VNC motor neurons between VB3 and VB8 are isolated from circuitry in both the head and tail, and the

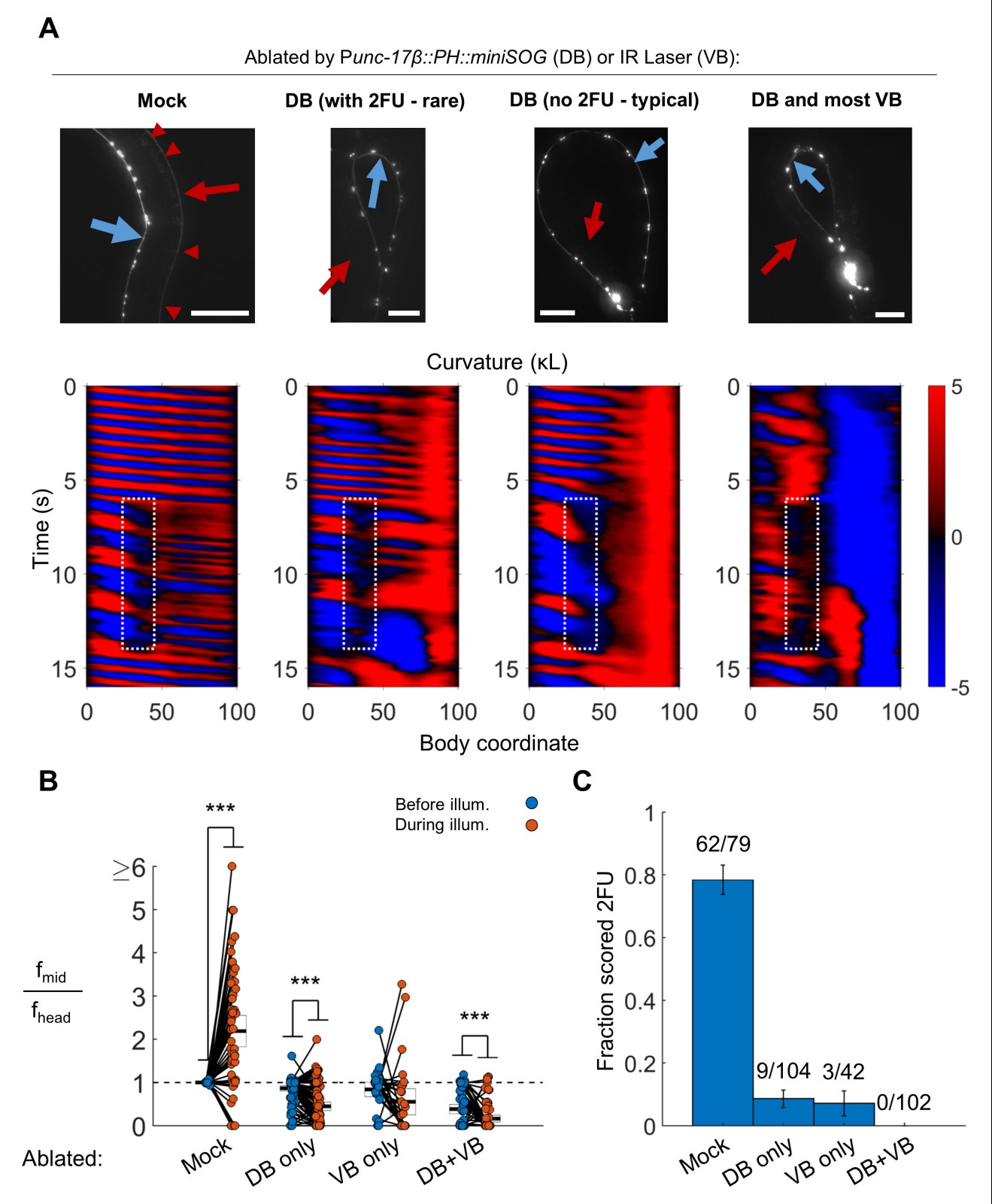

**Figure 5.** B motor neurons are required for 2FU. (**A**) Top panels: assessment of ablations using a P*acr-2::mCherry* label. After removal of dorsal B motor neurons (P*unc-17β::PH::miniSOG*), pairs of motor neurons – each corresponding to one VA and one VB type neuron – are visible along the VNC (blue arrows). Neither the DNC (red arrows) nor the DB or DA commissures (red arrowheads in mock) are visible. Scale bars: 100 μm. Bottom panels: Corresponding examples of worm locomotion after removal of DB (via miniSOG) and VB (via IR laser) motor neurons. Removal of DB always resulted in

*Figure 5 continued on next page*

*Figure 5 continued*

tail paralysis in a coiled position, but a minority of worms were able to generate a rhythmic midbody wave. Additional removal of VB motor neurons by laser ablation completely prevented 2FU and resulted in severe tail paralysis. (B,C) 2FU, as assayed by frequency measurements (B) or blinded, randomized scoring (C), is sharply reduced or eliminated by removal of DB, VB, or both. Head and mid-body frequencies were measured at body coordinates 15 and 60, respectively. Error boxes in (B) are the mean and 95% confidence interval of the mean. Error bars in (C) are standard error of the sample proportion. Numbers above each bar in (C) indicate the number of trials scored 2FU over the total number of trials for the condition; each individual worm contributed between one and five trials (3.6 on average). (***) p<0.001; paired t-test.

DOI: https://doi.org/10.7554/eLife.29913.016

The following figure supplement is available for figure 5:

**Figure supplement 1.** Behavior after ablation of VB motor neurons.

DOI: https://doi.org/10.7554/eLife.29913.017

VNC motor neurons posterior to VB8 are isolated from both the head circuits and the anterior VNC motor neurons. As before, these animals could generate robust body oscillations posterior to the first cut and higher frequency oscillations posterior to the second cut (*Figure 6C,E*, *Video 5*), suggesting that rhythms can arise independently from each of these portions of the VNC motor neurons. It should be noted that in all our VNC lesion studies, the severed processes of all premotor interneurons likely remained present in the VNC.

## B class motor neurons are necessary for rhythmic wave generation in VNC-lesioned worms

We next asked which motor neuron groups contributed to rhythm generation in worms in which the VNC and DNC were severed. The A motor neurons are not necessary for 2FU induced by neck paralysis (*Figure 4—figure supplement 1C*). We hypothesized that they are similarly not required for body oscillation in VNC-lesioned worms. We severed the VNC and DNC in either one or two locations after ablating the A motor neurons with P*unc-4::miniSOG*. Animals in which the VNC and DNC had been severed near the head were capable of robust wave generation and propagation posterior to the head (*Figure 6—figure supplement 1B*). Animals in which the VNC and DNC were severed in two locations were capable of rhythmic activity in the mid-body or tail, although we did not observe any cases of simultaneous oscillation in each segment (N = 20 worms, *Figure 6—figure supplement 1C*). These results support the idea that the A motor neurons are not required for generation of rhythmic forward waves in surgically isolated segments of the VNC motor circuit.

The B motor neurons were required for neck-paralysis-induced 2FU (*Figure 5*). We hypothesized that they are also required for forward undulatory rhythms in VNC-lesioned worms. To test this idea, we used VNC-lesioned worms in which the DA and DB motor neurons were ablated by miniSOG and the VB motor neurons were ablated by an infrared laser as before. After nerve cord surgery, worms in either B-ablation condition, like mock controls, were highly inactive. Application of a mechanical stimulus caused tight coiling, especially but not exclusively in the DB-only ablation condition (*Figure 6—figure supplement 2C*, *Video 6*). Despite this coiling, we observed at least one case of a VNC-lesioned, DB-ablated worm appearing to move the very tip of its tail, possibly in a rhythmic fashion, suggesting that DB neurons may not be required for rudimentary oscillation in at least the most posterior portion of the tail. However, in VNC-isolated worms for which most DB and VB neurons were ablated, we never observed rhythmic movements (*Figure 6—figure supplement 2D–F*; N = 19 mechanical stimulus trials from nine worms). These results suggest that, as was the case for optogenetic 2FU, the B motor neurons are required for rhythm generation in VNC-lesioned worms.

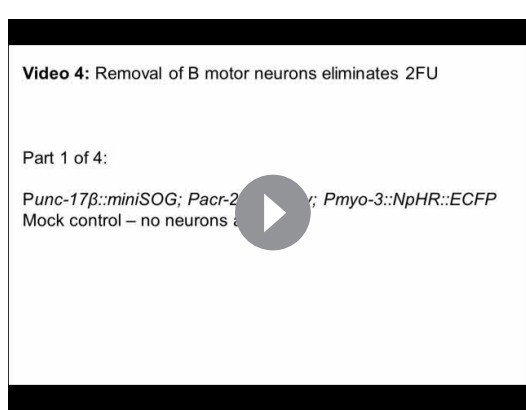

**Video 4.** Removal of B motor neurons by miniSOG and laser ablation eliminates 2FU.

DOI: https://doi.org/10.7554/eLife.29913.018

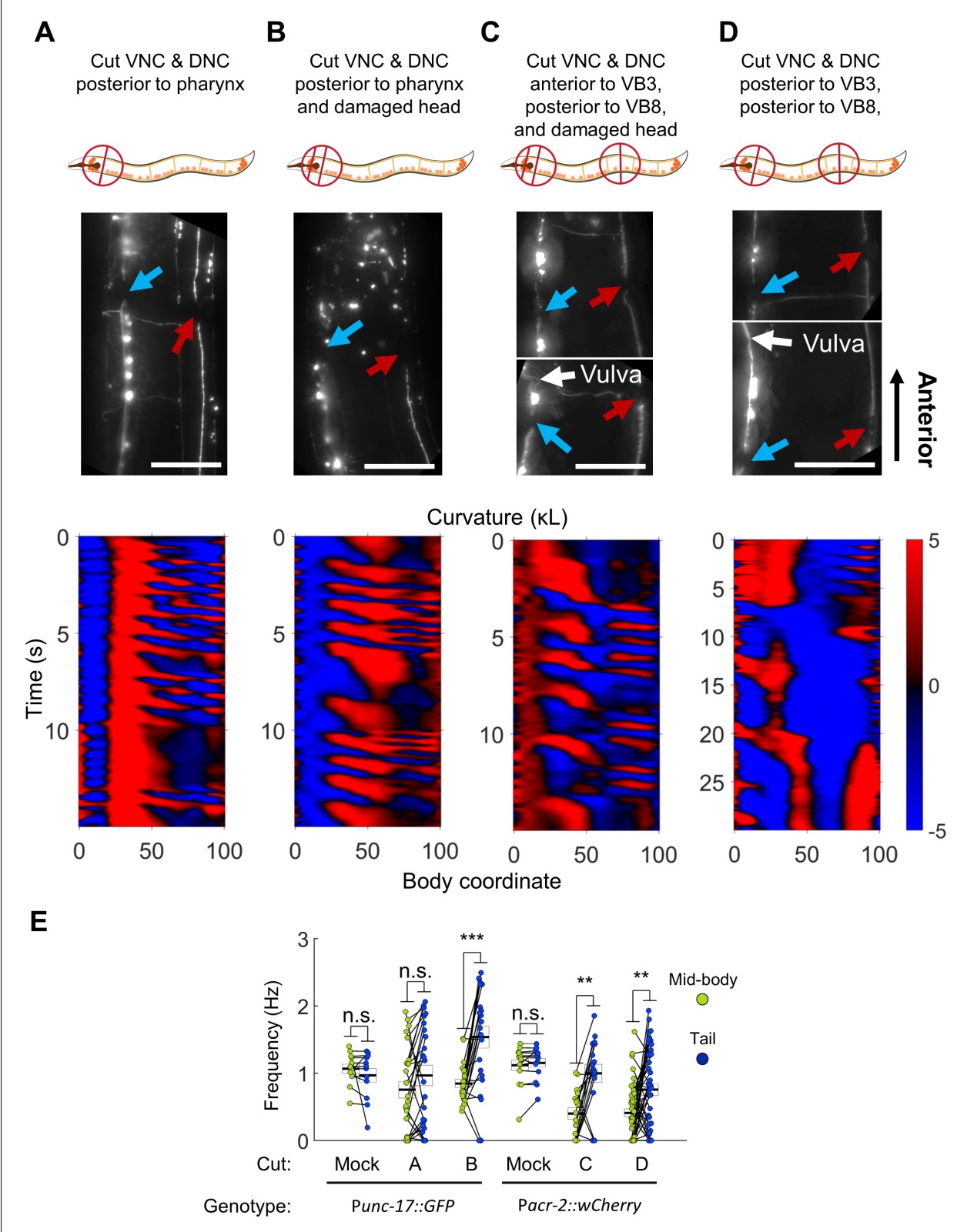

**Figure 6.** Undulations generated in the tail after severing the dorsal and ventral nerve cords. (**A**) The VNC (blue arrow) and DNC (red arrow) were severed in a P*unc-17::GFP* worm using a pulsed infrared laser. Several other dorso-ventral processes also appear cut. Nonetheless, robust bending waves are generated in the mid-body (lower pane). All scale bars: 50 µm. (**B**) The VNC and DNC are severed, and additional damage has been applied to the nerve ring to suppress head movements. Robust bending waves are generated in both the neck and tail (lower pane). (**C**) The VNC and DNC are

*Figure 6 continued on next page*

*Figure 6 continued*

severed posterior to the head and vulva (P*acr-2::wCherry*), and the nerve ring is lesioned to suppress head movements. Robust bending waves are generated in both the neck and tail (lower pane). (D) The VNC and DNC are severed posterior to the head and vulva, but the nerve ring was not targeted. Low-frequency head undulation and high-frequency tail undulation were observed separately in this animal. (E) Frequency of undulation in the mid-body and tail for all ablation conditions and mock controls. Each colored circle pair represents one bout of forward locomotion lasting at least 2 s. For each condition, data are only considered for worms in which the VNC and DNC are clearly severed in the indicated locations. Mid-body and tail frequencies were measured at body coordinates 45 and 85, respectively. Error boxes represent the mean and SEM. N = 3, 7, 3, 4, 5, and seven worms per condition, respectively. (*) p<0.05; (**) p<0.01; (***) p<0.001; paired t-test.

DOI: https://doi.org/10.7554/eLife.29913.019

The following figure supplements are available for figure 6:

**Figure supplement 1.** Body undulations after severing the VNC and DNC do not require the A motor neurons.

DOI: https://doi.org/10.7554/eLife.29913.020

**Figure supplement 2.** B motor neurons are required for body undulations after severing the VNC and DNC.

DOI: https://doi.org/10.7554/eLife.29913.021

The observation that anterior VNC/DNC cuts disrupt normal head undulation (*Figure 6A*) suggests that rhythm generation by the head circuit may require inputs from the VNC. To address this possibility, we studied additional worms in which the VNC/DNC were cut in two locations slightly more posterior to the head to reduce the likelihood of damage to the head motor neurons, but which were not subject to thermal damage to the head. As in our earlier experiment (*Figure 6A*), head movement was severely disrupted (not shown). However, we occasionally observed very slow head undulation in these animals, indicating that head undulation is still possible under these conditions (*Figure 6D*, *Video 5*).

One explanation for the low frequency of head undulations is that damage to the SMB or SMD neurons in the parallel tracts (*Figure 6A*) hampered head movement. Another possibility is that input from the posterior motor circuit is essential for the normal frequency of head undulation. The latter hypothesis may be supported by our earlier observation that strong decoupling of head and tail oscillations by muscle ablation (i.e. *without* lesioning the nerve cord or parallel tracts) resulted in similarly slow head oscillations (*Figure 3D*), while weak decoupling resulted in moderately slower head oscillations (*Figure 3A*). The possibility of posterior-to-anterior communication is discussed in detail in later sections.

These results suggest that the *C. elegans* forward locomotor circuitry possesses at least three units capable of independent oscillation, with two units located outside the head. Although we did not observe simultaneous three-frequency forward undulation in any animal tested, we have shown that oscillations are possible in each segment when isolated from the others.

## Undulations can arise in arbitrary portions of the VNC motor circuit

We next sought to identify the smallest portion of the VNC motor circuit that is capable of generating rhythmic behaviors, and whether any differences exist between the rhythmic properties of segments of various sizes. To address these questions, we again damaged the head (as in *Figure 6B*), then systematically varied the location of VNC/DNC lesions relative to the B motor neurons.

When we severed the VNC and DNC at any one of a number of different locations (anterior to VB3, VB5, VB6, VB7, VB9, VB10, and VB11), we found that undulations posterior to the damage arose near each cut location (*Figure 7B–E*). Bending amplitude similarly recovered just posterior to each cut location, rather than at a fixed body coordinate (*Figure 7F*). Oscillations in the

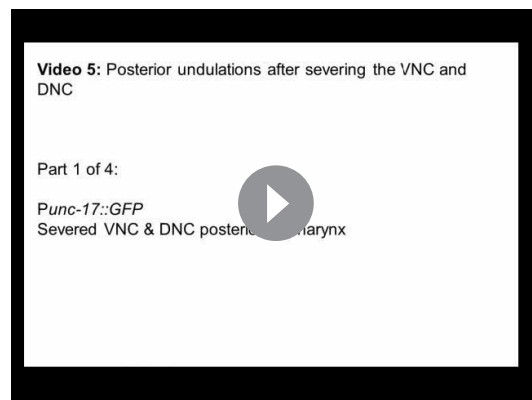

**Video 5.** Posterior undulations occur after severing the ventral and dorsal nerve cords with an infrared laser (VNC-isolated animals).

DOI: https://doi.org/10.7554/eLife.29913.022

tail usually had a higher frequency when the nerve cords were severed in the tail than when they were severed near the head (*Figure 7G*). We did not observe any bouts of locomotion with anterior-to-posterior waves posterior to the lesions when we severed the nerve cords anterior to VB11 (not shown), and bouts detected after lesioning at VB10 had waves of very low amplitude (*Figure 7F*), without clear rhythmicity (not shown). The smallest VNC segment that produced clear and robust rhythmic waves was the region between VB9 and the tail. These results suggest that the most compact rhythm-generating unit of the forward motor circuit is at least as small as the region containing VB9, VB10, VB11, DB6, and DB7.

## Rhythmic motor entrainment is possible in both the anteriorward and posteriorward directions

Although we have shown that the *C. elegans* forward motor circuitry contains multiple rhythm generating units, it remains unclear exactly how these oscillators are coupled together, or even if they are all active during normal locomotion. Previous work demonstrated proprioceptive coupling in the posterior direction (*Wen et al., 2012*), and we showed that a disruption to proprioceptive coupling, via optogenetic inhibition of neck muscles, could decouple body undulations from head movements (*Figures 3A* and *4*). One surprising feature of our results was that paralyzing the neck muscles appeared to decrease the head frequency. We found that during 2FU, head frequency decreases relative to the unperturbed swimming frequency (*Figure 8A*). Slowing was often even more dramatic when decoupling was stronger (*Figures 3D* and *6D*). These observations suggest that anteriorward coupling, in addition to posteriorward coupling, may be present in the forward locomotor circuitry.

To test for anteriorward coupling between motor circuit elements, we asked whether an oscillating optogenetic perturbation in the mid-body can entrain the head to a new frequency. We used our optogenetic targeting system to rhythmically inhibit the mid-body BWMs (*Figure 8B*). Worms subjected to this procedure exhibited a head bending frequency approximately one half that of the imposed frequency. The factor of one half is likely due to the presence of two phases during the rhythmic locomotory cycle of any single part of the body during which the curvature is close to zero (i.e. muscles are relaxed). In some cases, small head bends corresponding to individual mid-body pulses were evident as well (*Figure 8C*(i), *Video 7*).

We found the head frequency could be entrained to a range of imposed mid-body frequencies. Subjecting body coordinates 33–66 to pulsed illumination at frequencies from approximately 1 to 2 Hz caused an increase in the power spectrum of the worm's oscillations at frequencies corresponding to half of the imposed frequency, and a decrease at other frequencies. Pulsing at frequencies below 1 Hz generally led to head oscillations near the imposed frequency (*Figure 8C*(ii)). These results show that a mid-body rhythmic signal can entrain head bending, and point to the presence of an anteriorward coupling mechanism within the motor circuit.

We asked whether the anteriorward coupling occurs via the VNC cholinergic neurons, which are electrically coupled to each other, the muscle-to-muscle gap junctions, or through another mechanism. We applied a rhythmic optogenetic inhibition pattern to the midbody cholinergic neurons in P*unc-17::NpHR* worms. Once again, worms subject to this procedure quickly adjust their head bending frequency to match one-half of the imposed frequency (*Figure 8D*(i), *Video 7*). Moreover, rhythmic illumination of the tail cholinergic neurons at 2 Hz similarly increased the magnitude of head bending at 1 Hz (*Figure 8D*(ii)). Selective rhythmic hyperpolarization of the mid-body B motor neurons also sufficed to increase the magnitude of head bending at one-half of the stimulus frequency (*Figure 8—figure supplement 1A*), as did rhythmic hyperpolarization of the BWMs when muscle-to-muscle gap junctions were disrupted in only in the BWMs by a

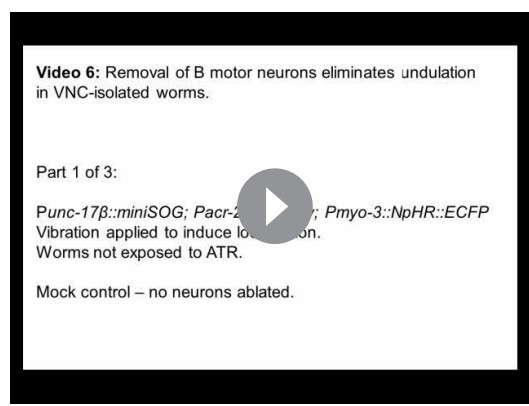

**Video 6.** Removal of B motor neurons by miniSOG and laser ablation eliminates undulations in VNC-isolated animals.
DOI: https://doi.org/10.7554/eLife.29913.023

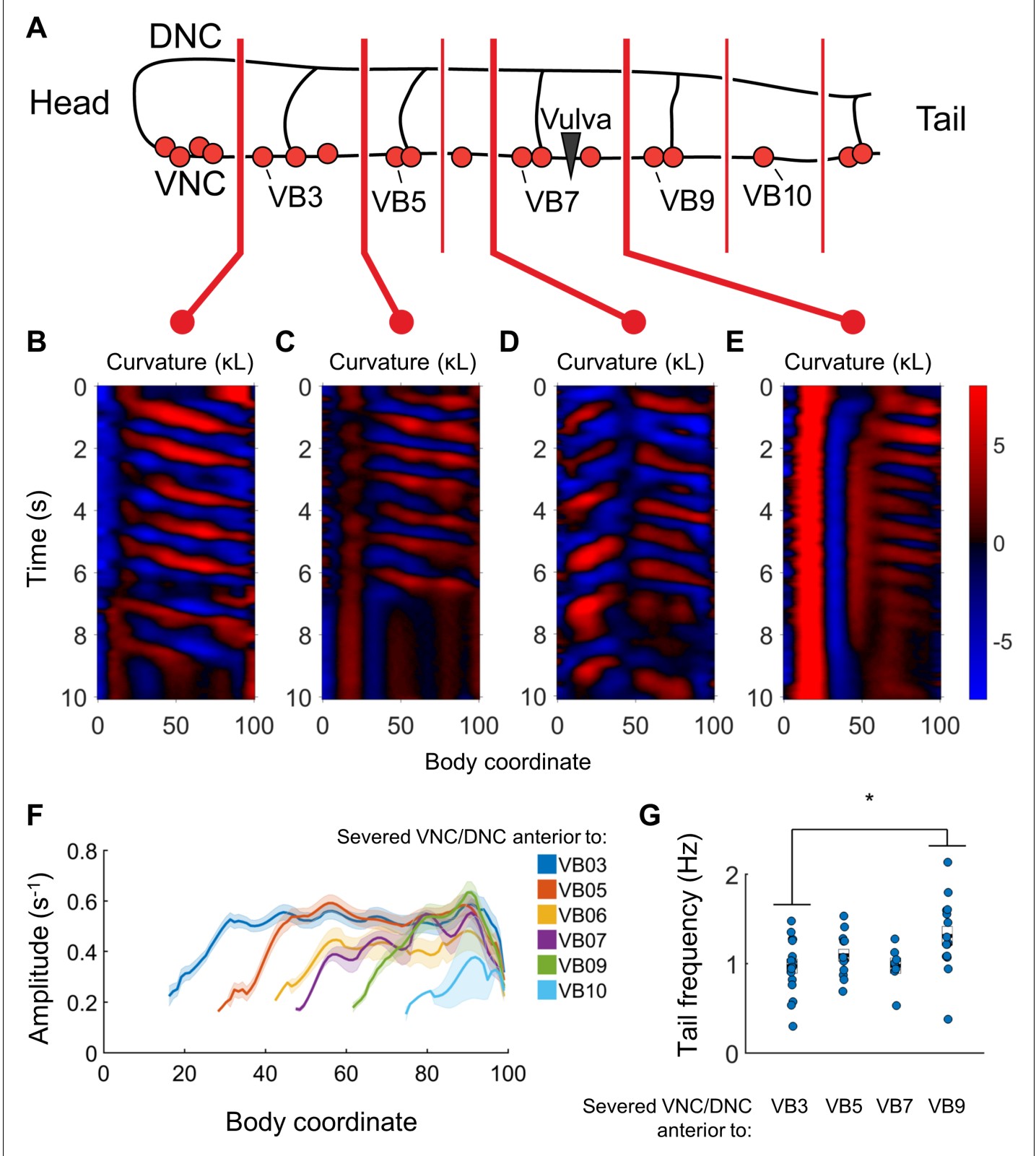

**Figure 7.** Undulations are generated after VNC/DNC lesioning in arbitrary locations. (**A**) Schematic indicating all regions at which we severed the VNC and DNC in relation to the B motor neurons. Each animal's nerve cords were severed at only one of these locations. The nerve ring of each worm was also damaged to restrict head movements as in **Figure 5B**. (**B–E**) Representative curvature maps for worms subject to four of the tested conditions. Note that anterior-to-posterior waves begin progressively more posterior for each cut location. In some cases, the head and tail exhibited waves

*Figure 7 continued on next page*

Figure 7 continued

propagating in opposite directions (D). (F) Amplitude of bending as a function of body coordinate after severing the VNC and DNC anterior to the indicated motor neuron. Only the portion of the curve posterior to the amplitude minima (the cut location) is shown. No bouts of locomotion with anterior-to-posterior waves were discernible posterior to the cut at VB11 either subjectively or by our analysis software. For cut locations at VB3, VB5, VB6, VB7, VB9, VB10, and VB11 we studied N = 15, 9, 6, 10, 14, 9, and 6 worms and observed 25, 16, 7, 12, 30, 19, and 14 bouts of forward locomotion (lasting at least 3 s), respectively. Shaded outline represents ±SEM. (G) Frequency of undulation at body coordinate 75 for four cut conditions. Boxes represent mean and SEM. Each colored circle indicates the frequency during one bout of forward locomotion. *p<0.05, one-way ANOVA with Bonferroni post-hoc comparisons.

DOI: https://doi.org/10.7554/eLife.29913.024

mutation in the innexin *unc-9* that was rescued in neurons but not muscles (*Wen et al., 2012*) (*Figure 8—figure supplement 1B*). These observations suggest that the posterior to anterior coupling is mediated neuronally, possibly by the VNC motor neurons.

## Discussion

In zebrafish and lampreys, rhythmogenic capability for swimming undulations is distributed along the rostro-caudal axis of the spinal cord (*Kiehn, 2006*; *Mullins et al., 2011*). When isolated from the rest of the cord, groups of lamprey spinal segments do not exhibit identical preferred frequencies (*Cohen, 1987*). In the swimming intact animal, oscillations in all segments are phase and frequency-locked by intersegmental coupling that spans broad swaths of the spine (*Mullins et al., 2011*).

The motor system of the leech, an invertebrate, also shows a distributed rhythm generating architecture. Individual ganglia of the leech VNC can generate crude bursting patterns that resemble their firing patterns during swimming. The most robustly oscillating ganglia are towards the middle of the leech's body, and isolated midbody ganglia also have a higher frequency than isolated ganglia near either the head or tail. In the intact animal, extensive, bidirectional intersegmental coupling drives the system to adopt a single locomotor frequency (*Pearce and Friesen, 1985*; *Kristan et al., 2005*).

Our results reveal a picture of forward locomotor control in *C. elegans* similar to that found in the lamprey and leech. We found that rhythmogenic capability in the worm is distributed along the VNC motor circuit. As in other swimming models, the rhythm-generating capability of posterior circuits is only detectable when coupling is disrupted. The rhythm-generating capability of posterior circuits was demonstrated in several ways: optogenetic inhibition of anterior neurons, optogenetic inhibition of anterior muscles, an inhomogeneous mechanical environment, or a lesion to the nerve cords. The incomplete nature of our optogenetic decoupling method yielded evidence that an anterior rhythm can entrain the higher frequency posterior rhythms. For example, we found that during 2FU, some but not all waves in the tail were continuous with waves in the head (*Figure 3B,C*). Even in these cases, the difference in locomotory frequency between the two body regions is inconsistent with single oscillator models.

We found that neither the head nor tail frequency during 2FU matched the natural (unperturbed) frequency of locomotion. Instead, the normal locomotory frequency in the environment of the assay was generally intermediate between the head and tail frequencies exhibited during 2FU. This finding shows another key similarity with models of swimming in other vertebrates and invertebrates: during locomotion, multiple rhythm generating units, each with different rhythmic properties, are combined by strong inter-unit coupling to form one functional unit (*Friesen and Hocker, 2001*; *Kristan et al., 2005*; *Kiehn, 2006*; *Mullins et al., 2011*). Such a whole-body oscillating unit will generally have rhythmic properties different from that of any subunit in isolation. Indeed, modeling studies of the leech swim CPG have suggested that the overall fictive locomotor frequency of a 17-ganglion portion of the VNC lies within the range of frequencies of the individual ganglia (*Zheng et al., 2007*). The gradient in intrinsic frequencies appears to set the wavelength during fictive swimming. The variation in rhythmic properties between different parts of the body may play a similar role in *C. elegans*.

During forward locomotion, the *C. elegans* motor circuit is coupled by posteriorward proprioception (*Wen et al., 2012*), and additional anteriorward and posteriorward coupling mechanisms (*Figure 8*). This bidirectional coupling appears to allow the entire circuit to operate as one unit. This

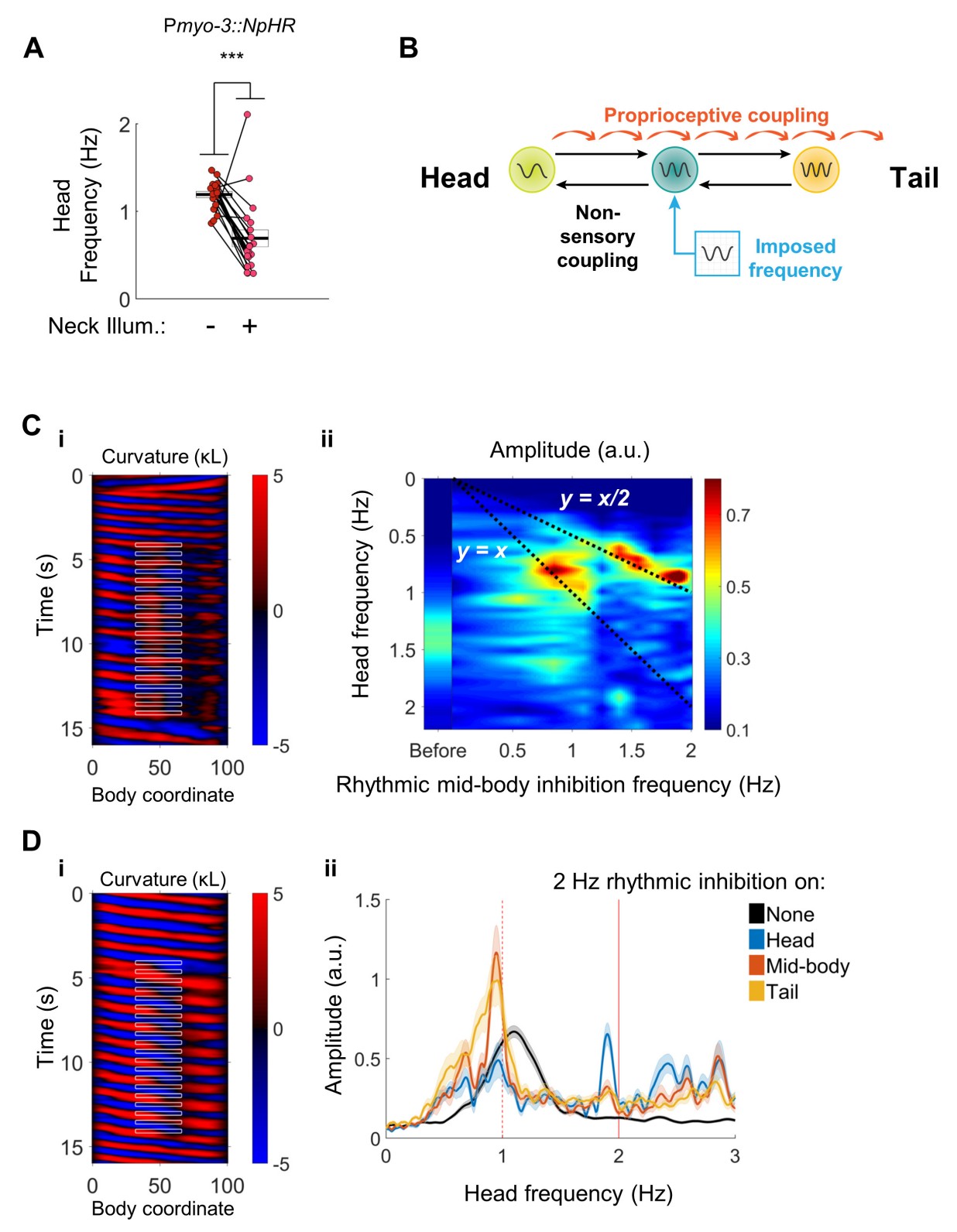

**Figure 8.** Head undulation frequency can be entrained by mid-body optogenetic manipulation. (**A**) Neck muscle hyperpolarization (*Figure 3A*) causes a significant decrease in head bending frequency. This decrease is not predicted by either model discussed in *Figure 1*. (**B**) A multi-oscillator model of forward locomotion allowing for motor coupling in both the anterior and posterior directions. To test this model, we sought to impose a new frequency on the mid-body of a freely moving worm, and test whether head bending also adopts the new frequency. (**C**) Head bending frequency can be

*Figure 8 continued on next page*

*Figure 8 continued*

entrained by rhythmically inhibiting the mid-body BWMs. (i) A curvature map showing a representative trial. Green light was pulsed on coordinates 33–66 at a frequency of 2 Hz onto a P*myo-3::NpHR* worm. Note that the head frequency slows to half of the imposed frequency, although some instances of a 1:1 correlation between a laser pulse and a head bend are also evident (e.g. around t = 13 s). (ii) Mean head frequency power spectra of P*myo-3:: NpHR* worms before manipulation (left bar, worms from all conditions are pooled) and while subject to rhythmic mid-body paralysis. Frequencies tested were 0 (with laser on), 0.5, 0.85, 1.1, 1.25, 1.4, 1.55, 1.7, 1.9, and 2.0 Hz. Frequency data are interpolated between these points. N ≥ 11 trials per condition, with each worm supplying at most two trials. For high-frequency inhibition (f > 1.1 Hz), the head is entrained to half of the inhibition frequency (bright peaks lie along y = x/2). For lower frequencies of inhibition (f ~ 0.85 Hz), the head is entrained to the inhibition frequency (bright peaks lie along y = x). (D) Head bending frequency can be entrained by rhythmically inhibiting the head, mid-body, or tail cholinergic neurons. (i) A curvature map showing a representative trial. Green light was pulsed on coordinates 33–66 at a frequency of 2 Hz onto a P*unc-17::NpHR* worm. (ii) Mean head frequency spectra before manipulation (black, all conditions pooled), and after rhythmically inhibiting the head (blue, body coordinates 0–33), mid-body (orange, 33–66), or tail (yellow, 66–99) neurons at 2 Hz. Rhythmic inhibition of the mid-body or tail increases the frequency power at 1 Hz and decreases the power at the original undulation frequency, mirroring (C). N ≥ 16 trials per condition, with each worm supplying at most two trials. Shaded outlines are the SEM. Vertical red lines indicate the imposed frequency (solid) or one-half of the imposed frequency (dashed).
DOI: https://doi.org/10.7554/eLife.29913.025

The following figure supplement is available for figure 8:

**Figure supplement 1.** Rhythmic activity in the mid-body B motor neurons is sufficient for posterior-to-anterior entrainment, and UNC-9 muscle-to-muscle gap junctions are not required.
DOI: https://doi.org/10.7554/eLife.29913.026

picture is similar to descriptions of leech and lamprey motor circuits. In the leech, oscillatory interneurons that comprise the swim CPG send axons along the anterior-posterior axis of the animal, and mediate bidirectional coupling between midbody ganglia (*Friesen and Hocker, 2001*; *Kristan et al., 2005*). The coupling between ganglia appears to result from a mixture of proprioceptive feedback and central control (*Yu et al., 1999*; *Kristan et al., 2005*). Injection of sinusoidally varying current into the leech stretch receptor cells was sufficient to entrain swimming activity (*Yu and Friesen, 2004*) at the injection frequency. Imposed rhythmic movements are also capable of entraining fictive swimming in isolated preparations of the lamprey spinal cord (*Grillner et al., 1981*; *McClellan and Sigvardt, 1988*) and several other vertebrate and invertebrate systems (*Wendler, 1974*; *Andersson et al., 1981*; *Robertson and Pearson, 1983*).

Our results include several lines of evidence implicating the cholinergic B and AS motor neurons as a likely source of posterior rhythmogenesis during *C. elegans* forward locomotion. First, disruption or removal of most or all B motor neurons virtually abolished independent tail undulations induced either optogenetically or by severing the nerve cords, but elimination of other motor neurons and premotor interneurons did not eliminate the ability of the posterior to oscillate independently (*Figures 4*, *5* and *6*). Second, stimulation of select B and AS neurons after paralyzing most of the body led to local, high-frequency oscillations in the tail that mimicked 2FU (*Figure 3—figure supplement 1*). Third, imposing a rhythmic signal on mid-body B motor neurons sufficed to entrain whole-body locomotor frequency, as did imposing rhythms on the broader class of cholinergic neurons and the body wall muscles (Figure 8). Changes in body posture are sensed by the B motor neurons (*Wen et al., 2012*), providing a likely explanation for how the BWM manipulation was able to entrain locomotion. The latter argument is somewhat comparable to one of the key criteria used to classify leech interneurons as part of the swim CPG; that injection of a pulse of current resets the phase of the locomotor rhythm (*Mullins et al., 2011*). In our experiments, the swimming frequency, and thus phase, was reset almost immediately after the first inhibitory stimulus, suggesting that the B motor neurons

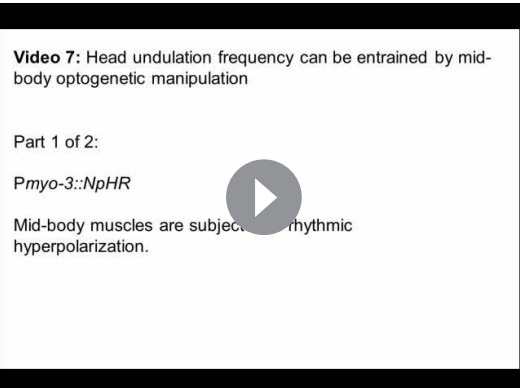

**Video 7.** Head undulation frequency can be entrained by rhythmic mid-body optogenetic manipulation of the muscles or cholinergic neurons.
DOI: https://doi.org/10.7554/eLife.29913.027

share the phase shifting property with oscillatory interneurons in the leech. Lastly, the B motor neurons are rhythmically active in phase with locomotion (*Kawano et al., 2011*; *Wen et al., 2012*), satisfying another key criterion set for candidate oscillator neurons in the leech (*Mullins et al., 2011*).

Descriptions of swimming CPGs in vertebrates have tended to exclude motor neurons as members of the CPG, with most oscillatory function attributed to interneurons (*Kiehn, 2016*). Leech excitatory motor neurons have not been shown to be members of the swim CPG, as injection of a pulse of current fails to reset the cycle phase (*Kristan et al., 2005*). However, leech inhibitory motor neurons are ascribed a role in swimming rhythm generation using the phase-resetting criterion (*Mullins et al., 2011*), motor neurons comprise the crab stomatogastric CPG (*Marder and Bucher, 2007*), and recent work has shown that motor neurons are key components of several locomotory pattern generators. In leech crawling, which consists of cyclic elongation and contraction phases, current pulses to the CV elongation motor neurons do indeed reset the phase of fictive crawling. However, the CV neurons were not concluded to be necessary components of the CPG because tonic hyperpolarization failed to abolish the crawling rhythm. Anatomical removal, arguably the more relevant test of necessity, was not reported (*Rotstein et al., 2017*). In zebrafish, motor neurons for swimming are bidirectionally coupled to locomotion-driving interneurons by gap junctions, and influence their recruitment, synaptic transmission, and firing frequency during locomotion (*Song et al., 2016*). Hence, there is a growing recognition that motor neurons are not limited to conveying oscillatory signals from interneurons, but may themselves participate in rhythm generation.

One difference between our results and previous findings in leeches is in the effect of severing the VNC. When we severed the VNC and DNC of *C. elegans*, we found that independent, generally higher frequency undulations occurred posterior to the severed region (*Figures 6* and *7*). Disruption of the leech VNC, by contrast, was not sufficient to prevent wave propagation from head to tail (*Yu et al., 1999*), suggesting that proprioceptive information suffices to propagate the wave. However, severing the VNC intersegmental coordinating neurons in in vitro preparations induced uncoordinated fictive swim oscillations at different frequencies occurring on either side of the cut (*Weeks, 1981*). This difference in results may arise due to differences between our thermal ablation method in *C. elegans* and physical severing of the leech VNC, or the relative span of proprioceptive signals in each system.

Our laser lesioning of the VNC likely did not remove the severed processes of premotor interneurons, nor did it prevent nonsynaptic neurotransmission, for example through neuropeptides, from potentially regulating rhythm generation across the lesion. These possibilities may account for the apparent difference in posterior rhythmogenic capability between worms in which AVB had been ablated versus severed. When the ventral nerve cord was isolated from the head ganglia, including the soma of AVB, rhythmic tail undulation was reliably evoked by a mechanical stimulus (*Figures 6* and *7*). However, independent tail undulations were observed only rarely after ablating AVB, even when the mechanical stimulus was applied (*Figure 4—figure supplement 1*). The AVB processes that likely remain in the severed VNC segments may continue to provide excitation to the B motor neurons to promote rhythm generation. The other apparent discrepancy, between disrupting AVB-B gap junctions by *unc-7* or *unc-9* genetic mutations and ablating AVB, may be a consequence of other effects of the mutation, including changes in other connections between the premotor interneurons and the B motor neurons (*Starich et al., 2009*), that may compensate for the loss of AVB:B gap junctions to activate the forward circuit. In any case, the observation that tail undulation was reliably evoked after eliminating all synaptic inputs from the head is inconsistent with the notion that independent oscillations from the tail require synaptic input from head circuitry.

Taken together, our results point to a new working model of *C. elegans* forward locomotion (*Figure 9*). Three oscillator units are depicted: an unknown head CPG, the VNC motor neurons between VB3 and AS7, and the VNC motor neurons between VB9 and AS11. The two VNC units are justified by our data from worms in which the VNC and DNC were cut in multiple locations (*Figure 6C*). We include AS in the model because it is the only class of VNC motor neurons that we have not directly investigated, and could be important for rhythm generation in the tail. The premotor interneurons, especially AVB, are important for activating the oscillatory circuit and may have additional unexplained roles in posterior rhythm generation. These are not the only circuit units capable of generating oscillations; when we severed the VNC and DNC at arbitrary locations, we found that oscillations resume closely posterior to each cut over a wide range of circuit sizes (*Figure 7*). Moreover, we

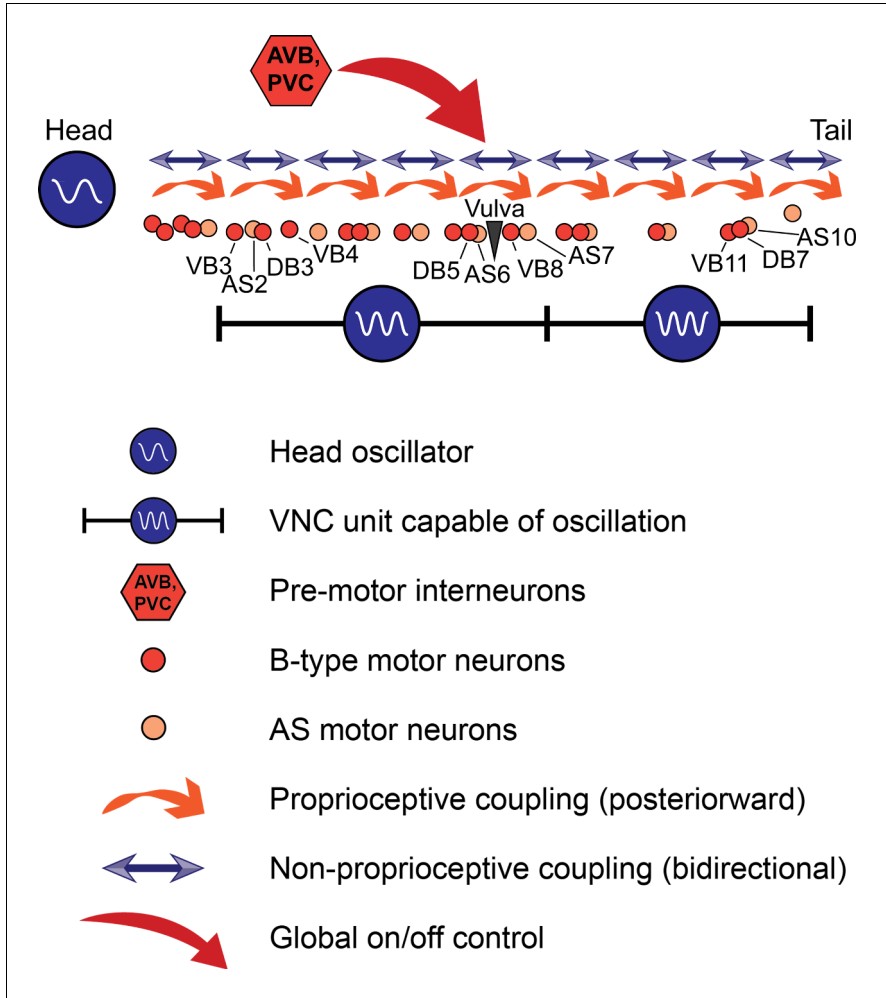

**Figure 9.** A model for *C. elegans* forward locomotion. Two units of the VNC motor neurons (and potentially more subsections) are capable of independent rhythm generation. However, all oscillating units are coupled by proprioceptive coupling (**Wen et al., 2012**) and another unknown, likely non-proprioceptive coupling mechanism that allows signaling in the anteriorward direction, and potentially also in the posteriorward direction. Pre-motor interneurons activate or suppress this circuit. AVB may have an additional, unexplained role in rhythm generation.
DOI: https://doi.org/10.7554/eLife.29913.028

cannot rule out the possibility that even smaller circuit units, perhaps even individual motor neurons, can generate rhythmic outputs.

Dissecting the relative contributions of cellular pacemakers, network oscillators, and reflex loops to rhythmic motor generation and coordination has been a longstanding challenge in vertebrates and invertebrates (**Kristan et al., 2005**; **Kiehn, 2006**). Our finding that the architecture for rhythm generation in *C. elegans* locomotion shares key properties with other vertebrate and invertebrate models sets the stage for molecular, cellular, and network-level investigations of motor coordination in a uniquely tractable model organism.

# Materials and methods

## Strains

We maintained *C. elegans* on 6 cm NGM plates seeded with *E. coli* OP50 at 20°C using standard methods (**Sulston and Hodgkin, 1988**). For all optogenetic experiments, we added 100 mM all-*trans* retinal (ATR) in ethanol at 0.8% by volume to the bacteria suspension before seeding the plates, and

kept plates in darkness. All strains were synchronized by hypochlorite bleaching and allowed to hatch on an NGM plate without food. L1 arrested larvae were transferred to OP50 or OP50 +ATR plates and allowed to grow to the appropriate stage. Unless otherwise specified, all experiments were performed using day 1 adult hermaphrodites.

All strains used in this study are listed in *Tables 1* and *2*. All transgenic strains were outcrossed a minimum of three times against N2.

## Construction and validation of the optogenetic targeting system

We constructed an optogenetic targeting system similar to that described in *Leifer et al., 2011* based around a Leica DMI4000B microscope. Dark-field illumination was provided by red LEDs and worms were imaged with a sCMOS camera (QImaging optiMOS). Custom written C++ software (*Fouad et al., 2017*) performs image segmentation of the worm and allows the user to select a region of the worm to illuminate. The illumination region is sent to a digital micromirror device (DMD), which then projects the laser onto only the desired portion of the worm through a 10X objective. The entire cycle runs at 40 Hz. A green laser (Shanghai Laser and Optics Century GL532T3-300, 532 nm wavelength, irradiance 10 mW/mm$^2$ at focal plane) was used for all activations of NpHR or Arch, and a blue laser (Shanghai Laser and Optics Century BL473T3-150, 473 nm wavelength, irradiance 4 mW/mm$^2$ at focal plane) was used for activation of ChR2. Worms were placed in a solution of 17% dextran in NGMB (*Fang-Yen et al., 2010*) in an 80-µm-thick chamber between a microscope slide and cover glass, separated by glass beads. NGMB is identical to NGM (*Stiernagle, 2006*) but without agar, peptone, or cholesterol.

To determine the accuracy of our illumination system, we studied worms expressing the excitatory opsin ChR2 under the control of the *aptf-1* promoter. These worms were reported to quickly become quiescent when the RIS head neuron is illuminated with blue light (*Turek et al., 2013*). We targeted a thin band of blue light to various locations along the centerline of each worm, and measured the amplitude of head bending waves as an output (data not shown). The distance between the region where most slowing occurred (body coordinate 14) and the nearest region with no slowing (body coordinate 20) suggests that our optogenetic targeting system has a resolution of about

**Table 1.** Transgenic arrays acquired or generated for this study

| Transgene | Plasmid or reference | Description | Purpose | Strain |
|---|---|---|---|---|
| vsIs48 | (*Chase et al., 2004*) | Punc-17::GFP | Identification of VNC and DNC | LX929 |
| akIs11 | (*Zheng et al., 1999*) | Pnmr-1::ICE; lin-15(+) | Ablation of INs | VM4770 |
| kyIs36 | (*Zheng et al., 1999*) | Pglr-1::ICE; lin-15(+) | Ablation of INs | VM4771 |
| wenIs001 | pJH2918 | Pacr-5::ArchT::RFP; lin-15(+) | Inhibition of B motor neurons | WEN001 |
| qhIs1 | (*Leifer et al., 2011*; *Husson et al., 2012*) | Pmyo-3::NpHR::ECFP; lin-15(+) | Inhibition of muscles | YX9 |
| qhIs2 | | | | YX10 |
| qhIs4 | pJH1841 | Pacr-2::wCherry; dpy-20(+) | Identification of B | YX146 |
| qhIs5 | (*Xu and Chisholm, 2016*) | Pmyo-3::PH::miniSOG(Q103L)+Pmyo-3::mCherry + Pttx-3::RFP | Ablation of BWM | YX203 |
| qhIs9 | (*Xu and Chisholm, 2016*) | Punc-17β::PH::miniSOG (Q103L)+Pacr-2::mCherry+ Pttx-3::RFP | Ablation of B | YX234 |
| hpEx803 | (*Wen et al., 2012*) | unc-9(fc16); hpIs3; Prgef-1-unc-9cDN)+odr-1 | Neuronal unc-9 rescue | ZM2509 |
| zxIs6 | (*Liewald et al., 2008*) | Punc-17::ChR2(H134R)::YFP;lin-15(+) | Stimulation of cholinergic neurons | ZX460/ ZM3265 |
| hpIs166 | (*Gao et al., 2015*) | Pglr-1::chop-2(H134R)::YFP; lin-15(+) | Identification of INs | ZM4624 |
| hpIs178 | (*Leifer et al., 2011*) | Punc-17::NpHR::ECFP; lin-15(+) | Inhibition of cholinergic neurons | ZM5016 |
| hpIs179 | (*Kawano et al., 2011*) | Psra-11::D3cpv | Identification of AVB | ZM5132 |
| hpIs366 | pJH2843 | Punc-4::tomm-20::miniSOG:: urSL::wCherry; lin-15(+) | Ablation of A and VC | ZM7690 |
| hpIs371 | | | | ZM7691 |

DOI: https://doi.org/10.7554/eLife.29913.029

**Table 2.** Additional strains generated by combining transgenes in *Table 1*.

| Strain | Transgenes | Description | Purpose |
|---|---|---|---|
| YX119 | qhIs1; unc-49(e407) | Muscle::NpHR, unc-49 | 2FU with D function impaired |
| YX126 | qhIs1; hpIs371 | Muscle::NpHR, A/VC::miniSOG | 2FU with A and VC removed |
| YX127 | hpIs178; hpIs371; zxIs6 | Cholinergic Neurons::NpHR&ChR2, A/VC::miniSOG | Inhibition or excitation of Cholinergic neurons with A removed |
| YX135 | qhIs1; vab-7(e1562) III. | Muscle::NpHR, DB disrupted | 2FU (fails) with DB disrupted |
| YX137 | qhIs1; unc-7(e5) | Muscle::NpHR, unc-7 | 2FU with AVB::B gap junctions disrupted |
| YX139 | qhIs1; unc-9(fc16); hpEx803 | Muscle::NpHR, UNC-9 disruption in muscles only | Entrainment with BWM::BWM gap junctions disrupted |
| YX140 | qhIs1; unc-9(fc16) | Muscle::NpHR, unc-9 | 2FU with AVB::B gap junctions disrupted |
| YX148 | qhIs1; qhIs4 | Muscle::NpHR, AB::RFP | 2FU with some B removed OR undulation with nerve cords severed |
| YX152 | hpIs166; akIs11 | IN::ICE&YFP | Assessment of ICE ablations |
| YX153 | hpIs166; kyIs36 | IN::ICE&YFP | Assessment of ICE ablations |
| YX159 | qhIs2; akIs11 | Muscle::NpHR, IN::ICE | 2FU with PVC removed |
| YX160 | qhIs2; kyIs36 | Muscle::NpHR, IN::ICE | 2FU with PVC removed |
| YX177 | hpIs366; vsIs48 | A/VC::miniSOG, Cholinergic Neurons::GFP | Undulation with nerve cords severed and A removed. |
| YX223 | qhIs1; qhIs9 | Muscle::NpHR, DB ablated | 2FU with B removed OR undulation with nerve cords severed and B removed. |
| YX230 | qhIs1; hpIs179 | Muscle::NpHR, AVB labeled for ablation | 2FU with AVB removed |

DOI: https://doi.org/10.7554/eLife.29913.030

6% of an adult worm's body length, similar to that reported for a previous system (*Leifer et al., 2011*).

## Optogenetic inhibition and stimulation

For head and neck optogenetic inhibition experiments, YX9 (Muscle::NpHR), ZM5016 (Cholinergic Neurons::NpHR), WEN001 (B::Arch), YX127 (Cholinergic Neurons::NpHR&ChR2, A/VC::miniSOG), YX119 (Muscle::NpHR, *unc-49(e407)*), YX135 (Muscle::NpHR, *vab-7(e1562)*, YX137 (Muscle::NpHR, *unc-7(e5)*), or YX140 (Muscle::NpHR, *unc-9(fc16)*) larvae were transferred to ATR plates and allow to grow to first day of adulthood. Up to 20 adult worms were mounted at a time on our optogenetic targeting system. Worms were sampled with replacement from the dextran chamber, and illuminated with a green (532 nm wavelength) laser a total of one to three times in the indicated region, with at least 10 s between successive illuminations of the same animal.

For activation of posterior B and AS motor neurons, YX127 (Cholinergic neurons::NpHR and ChR2, A/VC::miniSOG) worms were allowed to grow to the second larval stage on ATR plates, and then illuminated *en masse* with blue light (470 ± 17 nm wavelength) at 3 mW/mm$^2$ for 20 min to ablate the A-type motor neurons. Worms were transferred to a fresh ATR plate to grow for two more days. To confirm a loss of reversal capability in each day one adult, worms were individually prodded with a platinum wire worm pick. All worms tested failed to move backwards during this assay. Worms were then mounted on the optogenetic targeting system as before. Global amber (580 ± 15 nm wavelength) illumination was applied through the transmitted light port of the microscope and varied in irradiance until just strong enough to paralyze the worm, presumably through the action of P*unc-17::NpHR*. Once paralyzed or nearly paralyzed, 473 nm laser light was applied to the indicated region of the tail using the DMD.

For rhythmic inhibition of muscles or cholinergic neurons, synchronized YX148 (Muscle::NpHR, AB::RFP) L4 larvae, YX139 (Muscle::NpHR, BWM gap junctions disrupted) L4 larvae, YX127 day 1 adults, or WEN001, day 1 adults grown on ATR plates were mounted on the optogenetic targeting system and illuminated periodically at the indicated frequency and location through our custom software. For this experiment, L4 *qhIs1* larvae were used because expression of *myo-3::NpHR::ECFP* in the BWMs appeared to be weaker in adults. YX127 animals were not exposed to blue light prior to this experiment.

## Neuron and muscle photoablation using miniSOG

For optogenetic 2FU experiments, YX126 (Muscle::NpHR, A/VC::miniSOG) larvae were allowed to grow for 2 days on OP50 plates until the L4 stage, at which RFP was visible in the both the A- and VC-type motor neurons. Worms were bulk illuminated with blue light (wavelength 470 ± 17 nm) at 3.5 mW/mm$^2$ for 20 min of total illumination time using 0.5 s on/1.5 s off pulse train (*Qi et al., 2012*) from a Leica EL6000 light source. After illumination, the larvae were transferred to OP50+ ATR plates and incubated for two additional days, but did not appear to grow past the L4 stage. During the optogenetic experiments, worms were observed swimming forward and stopping, but never in reverse. In addition, worms were handled individually and recovered from the dextran chamber for fluorescence imaging. We mounted each worm on a separate 4% agar pad and acquired RFP images on a compound fluorescence microscope (Leica DMI6000B). In control worms, which grew into adults, RFP was visible in the VC-type motor neurons, some A-type motor neurons (RFP was visible in all A-type motor neurons at the L2-L4 stages), and the posterior intestine. The normal RFP expression pattern in A- and VC-type motor neuron was absent in all illuminated worms. However, neurons in the tail corresponding to VA12, DA8, or DA9 were usually visible (see also *Figure 6—figure supplement 1A*).

For VNC/DNC cauterization experiments, YX177 (AB::RFP, A/VC::miniSOG) worms were exposed to blue light at the L2 stage using 0.5 s on/1.5 s off pulsing and surgically manipulated at the day 2 adult stage as described below. Ablation of the A motor neurons was assessed by recording GFP fluorescence images of each animal after behavioral imaging. In all animals, only B and AS motor neurons were visible along most of the VNC, with the exception of the three posterior A motor neurons noted above.

For body wall muscle ablation experiments, YX203 (BWM::miniSOG) adults were immobilized on agar pads using polystyrene beads (*Kim et al., 2013*) and the indicated region of the body was exposed to light with wavelength 470 ± 20 nm at an irradiance of 75 mW/mm$^2$ through a 40x objective on a Leica DMI6000B or DMI4000B microscope for 5.5 min with a 0.5 s on/0.5 s off pulse protocol (*Xu and Chisholm, 2016*). Spatial selectivity was achieved by restricting the diameter of illumination by adjusting the microscope's epifluorescence field diaphragm. After illumination, worms were recovered to an unseeded plate and immediately transferred to a 17% dextran chamber for behavioral imaging.

For B ablation experiments using P*unc-17β::PH::miniSOG* (YX223), we found that illumination at any larval stage preferentially killed the DA and DB motor neurons, but left most VA and VB alive, as determined by the loss of all dorsal commissures and DNC labeling by P*acr-2::wCherry* (not shown). Hence, we illuminated all YX223 animals at the L1 stage for 20 min with 0.25 s on/0.25 s off pulsing (*Xu and Chisholm, 2016*) to kill DB. Worms were recovered to ATR (for optogenetic 2FU) or regular seeded (for VNC surgery) plates. Additional laser ablation of VB neurons was performed at the L4 stage, and lesioning of the VNC and DNC was performed at the day 2 adult stage (both described below).

## Ablation of premotor interneurons by ICE

Prior to conducting optogenetic experiments, we generated strains YX152 (P*nmr-1::ICE*; P*glr-1::ChR2::YFP*) and YX153 (P*glr-1::ICE*; P*glr-1::ChR2::YFP*) to test whether the interneurons were appropriately ablated. Control ZM4624 (P*glr-1::ChR2::YFP*) L1 larvae and adults showed bright YFP labeling of many neurons, including many head neurons and PVC, the only pair of labeled neurons in the tail.

In L1 arrested YX152 and YX153 worms, many neurons were also easily visualized by YFP, although normal locomotion was impaired. By the adult stage, YFP signals in all head and tail neurons had nearly vanished in all YX153 worms (N = 18); some small and faint fluorescent puncta, similar in appearance to intestinal birefringent granules, were visible in the nerve ring and tail. These dim puncta did not have visible processes, and we were unable to identify any head neurons or PVC in any of these worms. In YX152 adults, PVC was similarly not identifiable in any worm (N = 10), although many brightly YFP-labeled neurons were visible in the nerve ring. These results suggest that all or most P*glr-1* positive interneurons are ablated in P*glr-1*::ICE worms and that PVC (and likely other P*nmr-1* positive neurons) are ablated in P*nmr-1*::ICE worms. AVB are likely present in both P*glr-1*::ICE and P*nmr-1*::ICE worms. (*Kawano et al., 2011*); Kawano, Po, and Zhen, unpublished).

For optogenetic experiments, we generated strains YX159 (Muscle::NpHR, IN::ICE) and YX160 (Muscle::NpHR, IN::ICE) and performed optogenetic illuminations as in our original optogenetic 2FU experiments.

## Pulsed infrared laser cauterization of neurons and nerve cords

For ablation of B motor neurons or AVB interneurons, YX148 (Muscle::NpHR, AB::RFP), YX223 (Muscle::NpHR, DB killed by miniSOG, see above), or YX230 (Muscle::NpHR, AVB labeled) worms were raised on ATR plates until most animals were at the L3 to L4 stage, and then immobilized on 4–10% agar pads using 50 nm polystyrene beads (*Kim et al., 2013*). Each pad was mounted on a pulsed infrared laser system (*Churgin et al., 2013*) that had been modified with increased power to deliberately kill cells. Each neuron, visualized by RFP or GFP fluorescence optics, was irradiated with a single 0.8 to 1.6 ms pulse of the 400 mW laser through a 63X oil immersion objective. We determined that a single 0.8 ms pulse reliably kills a targeted VNC motor neuron, has a 50% chance of killing a VNC neighbor within 2.5 µm, and has a 10% chance of killing a VNC neighbor within 5 µm from the target (A. D. F. and C. F.-Y., unpublished data). After ablation, worms were recovered and transferred to a fresh ATR plate to resume development for one additional day. During optogenetic experiments, worms were handled individually and recovered to agar pads after illumination.

To sever the VNC and DNC, day 2 adult YX148 or LX929 (Cholinergic neuron::GFP) worms were immobilized with polystyrene beads and mounted on our infrared laser system as before, and the indicated area of the wCherry- or GFP-labeled cord was illuminated with a train of 10–25 IR laser pulses with 2 ms duration. Worms were transferred to fresh unseeded plates and allowed to recover for at least 4 hr before behavioral and fluorescence imaging. For behavioral imaging, worms were mounted individually in 17% dextran chambers and recorded swimming for at least 1 min under dark-field illumination. Most worms were inactive 4 hr after surgery, especially when the VNC and DNC were lesioned in two locations (not shown).

In many other systems, mechanical, electrical, or chemical stimuli can be applied to induce swimming or fictive swimming in an otherwise quiescent preparation (*Kristan et al., 2005*). To agitate *C. elegans*, we mechanically vibrated each worm using a 200 Hz cell phone motor for periods of 10–20 s to induce locomotion at least twice during each recording. After behavioral imaging, each worm was transferred to a pad and imaged for red or green fluorescence imaging.

## Fluorescence imaging and identification of neurons and nerve cord lesions

For all neuron ablation or nerve cord lesioning experiments, we acquired RFP or GFP fluorescence images of each animal through a 40X objective on a compound fluorescence microscope (Leica DMI 6000B). Through examination of images, A and B type motor neurons or AVB interneurons were labeled as present or missing based on the location, stereotypic ordering, commissural orientation, and the presence or absence of commissures from dorsal A- or B-type motor neurons. In ablation conditions labeled 'most VB' in *Figure 5* and *Figure 6—figure supplement 2*, we only used data from animals in which at most three out of the nine VB neurons between VB3 and VB11 inclusive were visible. In DB ablation conditions in the same two figures, all DB motor neurons appeared to be missing, and occasionally some VA/VB were missing as well. However, we generally could not identify B motor neurons anterior to VB3 because of clustering and the presence of very bright AIY:: RFP in *qhIs9* worms. In each category in *Figure 4F*, we included only worms for which all indicated B motor neurons were missing. Some individuals in each category had additional missing neurons. In *Figure 4—figure supplements 3* and *4*, all ablated B motor neurons are indicated for each individual worm example. In AVB ablation conditions, we only analyzed data from worms in which both AVB cell bodies and their associated processes were removed. Because cell killing occurs over a ~ 3 µm radius volume in our system, it is highly likely that other head neurons were also damaged or killed by this procedure. For nerve cord lesioning experiments, we only analyzed data from animals in which all indicated VNC/DNC targets were clearly severed.

## Head lesions using a heated wire

To broadly lesion the head and inhibit anterior bending, four freely crawling adult N2 worms were gently touched on or near the head with a platinum wire attached to a soldering iron. The worms

appeared to crawl backwards after the initial touch, so we applied a second touch to the agar near the tail to induce forward locomotion. We recorded behavior immediately after lesioning.

### Heterogenous mechanical environment experiments

Day 1 adult N2 worms were transferred to a slide containing 3 to 5 µL islands of solutions of high viscosity 3% hydroxypropylmethylcellulose (HPMC, Ashland Benecel K200M) in NGMB, surrounded by NGMB without HPMC. A second slide, spaced by 125 µm thick plastic spacers, was placed on top to form a two-dimensional chamber similar to those used for optogenetics experiments, but with an inhomogeneous mechanical environment. We imaged each slide under dark field illumination on a Nikon TE2000-S microscope and recorded worms transitioning from low-viscosity to high-viscosity regions.

### Curvature segmentation, analysis, and statistics

For experiments with our optogenetic targeting system, the real-time segmentation for body targeting was recorded to disk along with each video frame. We wrote custom MATLAB codes (*Fouad et al., 2017*) to compute the curvature of the worm in each frame using the recorded centerline coordinates. All analysis codes, codes for the optogenetic targeting system, and source data for figures are freely available (*Fouad et al., 2017*). Frequencies and amplitudes in optogenetic experiments were measured over either a 3 s non-illuminated interval ending at the start of illumination, or a 3 s illuminated interval beginning at the start of illumination. Trials were excluded if, during the period of analysis, (1) the worm showed reverse locomotion at any time, or (2) the segmentation algorithm was disrupted, for example if the worm touched a bubble, another worm, or the edge of the chamber.

For all other experiments, worm segmentations were generated from dark-field videos using WormLab software (MBF Bioscience, Williston, VT). The centerline coordinates were exported and curvature maps constructed as before. To identify bouts of forward locomotion in 1–2 min worm recordings, we computed the activity level and wave direction in the kymogram as a function of time and body coordinate. Bouts of forward locomotion in body segments were identified when the activity level was higher than a fixed threshold and the local direction of wave propagation was anterior-to-posterior for longer than the amount of time specified (typically 2–3 s).

To measure frequencies of undulation at any body coordinate, we computed the Fourier transform of time derivative of the curvature, and identified the frequency corresponding to the maximum amplitude within a 0 to 2.5 Hz window. For trials in which no clear peak emerged (i.e. the maximum amplitude was less than a fixed threshold), no undulations were considered to have occurred and the frequency was treated as zero. The same threshold was used for every frequency measurement presented. In ratiometric measurements, a small number (<2%) of ratios with infinite values were excluded. Bending amplitudes were calculated as the root mean square of time differentiated curvature. Changes in the mean frequency, frequency ratio, or bending amplitude of all paired data were evaluated using a paired t-test.

For blinded, randomized scoring of trials shown in *Figure 5* and *Figure 6—figure supplement 2*, kymograms were scored manually without prior examination of the data. Trials were scored as '2FU' or 'oscillating' if the following criteria were met: (1) bending waves were visible in the tail that did not appear to arise from the head, and (2) at least two complete undulatory cycles occurred in the tail, in any portion of the optogenetic inhibition or vibration stimulus time windows, which were 8 and 20 s, respectively. Trials were marked for exclusion if the windows contained exclusively reversal or incorrectly segmented behavior. The manual scoring method allowed detection of 2FU occurring outside the 3 s window used for quantitative analysis, which is why more 2FU trials were found in the DB ablation condition by manual scoring than by quantitative frequency analysis.

## Acknowledgements

We thank Mei Zhen, Quan Wen, Min Wu, Michelle Po, Yishi Jin, Andres Villu Mariq, and Alexander Gottschalk for providing strains. Some strains were provided by the by the CGC, which is funded by NIH Office of Research Infrastructure Programs (P40 OD010440). CF-Y was supported by the National Institutes of Health, Ellison Medical Foundation, and Sloan Research Foundation. ADF was supported by the National Institutes of Health. ST was supported by an Abraham Noordergraaf

Research Fellowship and a Littlejohn Fellowship. JRM was supported by a Holtz Undergraduate Research Fellowship. We thank Mei Zhen, Michael Nusbaum, David Raizen, Vijay Balasubramanian, Robert Kalb, Gal Haspel, Brian Chow, and Edward Fouad for helpful suggestions and discussions, Matthew Churgin for technical assistance, and Wassana Techadilok for assistance with figure preparation.

## Additional information

### Funding

| Funder | Grant reference number | Author |
|---|---|---|
| National Institutes of Health | R01 NS-084835-05 | Anthony D Fouad<br>Hongfei Ji<br>Christopher Fang-Yen |
| University of Pennsylvania | Abraham Noordergraaf Research Fellowship | Shelly Teng |
| University of Pennsylvania | Littlejohn Fellowship | Shelly Teng |
| University of Pennsylvania | Holtz Undergraduate Research Fellowship | Julian R Mark |
| Canadian Institutes of Health Research | | Asuka Guan |
| Lawrence Ellison Foundation | New Scholar in Aging Award | Christopher Fang-Yen |
| Alfred P. Sloan Foundation | Sloan Fellowship | Christopher Fang-Yen |

The funders had no role in study design, data collection and interpretation, or the decision to submit the work for publication.

### Author contributions

Anthony D Fouad, Conceptualization, Data curation, Software, Formal analysis, Supervision, Investigation, Visualization, Methodology, Writing—original draft, Project administration, Writing—review and editing; Shelly Teng, Julian R Mark, Alice Liu, Pilar Alvarez-Illera, Hongfei Ji, Angelica Du, Priya D Bhirgoo, Eli Cornblath, Investigation; Sihui Asuka Guan, Resources; Christopher Fang-Yen, Conceptualization, Software, Formal analysis, Supervision, Funding acquisition, Investigation, Writing—original draft, Writing—review and editing

### Author ORCIDs

Anthony D Fouad [iD] http://orcid.org/0000-0002-4677-2968
Christopher Fang-Yen [iD] https://orcid.org/0000-0002-4568-3218

### Decision letter and Author response

Decision letter https://doi.org/10.7554/eLife.29913.035
Author response https://doi.org/10.7554/eLife.29913.036

## Additional files

### Supplementary files

• Transparent reporting form
DOI: https://doi.org/10.7554/eLife.29913.031

### Major datasets

The following dataset was generated:

| Author(s) | Year | Dataset title | Dataset URL | Database, license, and accessibility information |
|---|---|---|---|---|
| Anthony D Fouad, Shelly Teng, Julian R Mark, Alice Liu, Pilar Alvarez-Illera, Hongfei Ji, Angelica Du, Priya D Bhirgoo, Eli Cornblath, Asuka Guan, Christopher Fang-Yen | 2017 | Data from: Distributed rhythm generators underlie *Caenorhabditis elegans* forward locomotion | https://doi.org/10.5061/dryad.q0d1n | Available at Dryad Digital Repository under a CC0 Public Domain Dedication |

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
