## [Decision Letter]

Thank you for submitting your article "Distributed rhythm generators underlie *Caenorhabditis elegans* forward locomotion" for consideration by *eLife*. Your article has been favorably evaluated by Eve Marder (Senior Editor) and three reviewers, one of whom, Ronald L Calabrese (Reviewer #1), is a member of our Board of Reviewing Editors. The following individual involved in review of your submission has agreed to reveal their identity: David M Miller III (Reviewer #3).

The reviewers have discussed the reviews with one another and the Reviewing Editor has drafted this decision to help you prepare a revised submission.

Summary:

This is a very interesting manuscript that is very important for the *C. elegans* model system, and has general interest as well. It dissects the motor system of *C. elegans* by targeted optogenetic manipulation, laser ablation experiments, and mutant analysis and uses behavior as assays to identify and define the forward locomotion "CPG". While CPG may not be the right term in the minds of many, the authors do indeed identify sources of oscillation in motor neurons that underlie rhythmic forward locomotion.

Multiple sections of forward locomotor circuitry (VNC) are capable of independently generating undulatory body rhythms. The driver of these multiple rhythm generators is localized to cholinergic motor neurons in the midbody, mainly B-MNs dedicated to forward locomotion. Using rhythmic optogenetic perturbation, bidirectional entrainment between different sections of the body was revealed. The distributed nature of the network and the bidirectional nature of the coordination mechanisms make the system analogous to swimming in vertebrates such as lamprey and zebra fish which broadens interest. The work is very careful and systematic with appropriate controls, and the data analyzed appropriately. Necessary data appear in figures in an easily accessible form.

Essential revisions:

1) The Discussion and Introduction need major reconsideration. The concept of CPG is not well articulated and its meaning is stretched, and is not actually consistent with much of the literature in other animals. You are defining units of "oscillation" which map to some extent onto B-MNs. You are at the beginning of getting down to the oscillators and networks such as may exist. Equating the structure of the forward locomotion circuit in *C. elegans* with mammalian spinal cord in limbed mammals seems unwarranted. You may be dealing with something reminiscent of vertebrates like lamprey and fish which use axial locomotion (swimming) or invertebrate networks like that for leech swimming and crawling *but* you have not identified the role of the basic antagonisms at all. What causes dorsal-ventral alternation and can one rhythm exist without the other? You should drop the insistent comparison to limbed mammals (or at least read the literature of the past decade); each limb has its own CPG and there are two limb enlargements in the SC where these reside. Phasing between the limb CPGs is variable (gaits). You should read up on zebra fish and lamprey and also other inverts. Leech crawling (Szczupak) has some things in common with what you have discovered as does leech swimming. You have a tremendous quantity of high quality data that advances work in your area strongly but overselling it to the mammalian community does not serve you well.

2) In a related preprint released by some of the same authors, "A descending pathway facilitates undulatory wave propagation in *Caenorhabditis elegans* through gap junctions" (https://doi.org/10.1101/131490), there appear to be discrepancies in the results which would be useful to clarify. In particular, in Figure 6 of the preprint, inhibition of mid-body B-type motor neurons appears to disrupt both head and tail oscillators while Figure 3 of the manuscript show disruption only of the tail. Figure 3 of the preprint also appears to show different results.

3) Subsection “Several classes of motor neurons are not required for forward locomotion or 2FU”. The conclusion that B and/or AS type cholinergic neurons are required for rhythmic oscillations depends on the finding that direct ablation of other classes of ventral cord motor neurons (A, D, VC) does not. Could this question be addressed in part by ablating all B-neurons with *acr-5::mini::SOG*? Similarly, the *unc-55* promoter is selectively expressed in VD and AS neurons and could therefore be used to determine if AS neurons are required since DD + VD function is not. The *unc-53* promoter, expressed in DA + AS neurons, could also be used for this experiment since DA neurons are not required. The authors should consider these new experiments, which may be quite manageable in a short time, or provide a strong rationale for why they are not appropriate.

4) Experiments described here either silence potential local sources of oscillation or sever physical connections between motor neurons and distal potential oscillators to conclude that motor circuit oscillations likely originate from distributed foci in the axial nerve cord. These experiments do not rule out the possibility, however, of "wireless" regulators potentially involving neuropeptide signals. The authors should address this point in the Discussion.

5) Subsection “Premotor interneurons are not essential for forward movement or 2FU”, last paragraph uses genetic mutants to eliminate gap junctions composed of the innexins UNC-7 and UNC-9 between AVB and B-type motor neurons to argue that the loss of AVB to B-type motor neuron electrical synapses does not prevent independent oscillations of anterior vs posterior body regions. This result is over-interpreted because *unc-7* and *unc-9* mutants would not eliminate other potential AVB B-motor neuron gap junctions composed of other innexins.

---

## [Author Response]

Essential revisions:1) The Discussion and Introduction need major reconsideration. The concept of CPG is not well articulated and its meaning is stretched, and is not actually consistent with much of the literature in other animals. You are defining units of "oscillation" which map to some extent onto B-MNs.

We have extensively revised the Introduction and Discussion. As much as possible, we sought to avoid the term CPG and instead refer more generally to rhythm generators, which do not necessarily satisfy the criteria of classical CPGs.

You are at the beginning of getting down to the oscillators and networks such as may exist. Equating the structure of the forward locomotion circuit in C. elegans with mammalian spinal cord in limbed mammals seems unwarranted. You may be dealing with something reminiscent of vertebrates like lamprey and fish which use axial locomotion (swimming) or invertebrate networks like that for leech swimming and crawling but you have not identified the role of the basic antagonisms at all. What causes dorsal-ventral alternation and can one rhythm exist without the other? You should drop the insistent comparison to limbed mammals (or at least read the literature of the past decade); each limb has its own CPG and there are two limb enlargements in the SC where these reside. Phasing between the limb CPGs is variable (gaits). You should read up on zebra fish and lamprey and also other inverts. Leech crawling (Szczupak) has some things in common with what you have discovered as does leech swimming. You have a tremendous quantity of high quality data that advances work in your area strongly but overselling it to the mammalian community does not serve you well.

As suggested, we have revised the Introduction and Discussion to focus on swimming organisms. We have added extensive discussions of comparisons of our results with those from leech, lamprey, and other systems, and minimized comparisons with limbed locomotion.

2) In a related preprint released by some of the same authors, "A descending pathway facilitates undulatory wave propagation in Caenorhabditis elegans through gap junctions" (https://doi.org/10.1101/131490), there appear to be discrepancies in the results which would be useful to clarify. In particular, in Figure 6 of the preprint, inhibition of mid-body B-type motor neurons appears to disrupt both head and tail oscillators while Figure 3 of the manuscript show disruption only of the tail. Figure 3 of the preprint also appears to show different results.

We believe these differences arise from differences in the amount of inhibition applied. We found that inhibition of mid-body cholinergic or specifically mid-body B-type motor neurons can have local or global disruptive effects on locomotory behavior, depending on the strength of inhibition. The strength of the optogenetic inhibition depends on laser power, laser wavelength (to our knowledge, the preprint cited above used a yellow laser which activates Arch more efficiently than our green laser), concentration and freshness of the all-trans-retinal cofactor, size of the region illuminated, and other conditions.

We are in agreement with results by Xu et al. that strong inhibition of the mid-body motor circuit can disrupt head movement, but we did not focus on this point because (1) it is a central point of the Xu et al. paper, and (2) we showed such anteriorward coupling via our bidirectional entrainment experiment.

3) Subsection “Several classes of motor neurons are not required for forward locomotion or 2FU”. The conclusion that B and/or AS type cholinergic neurons are required for rhythmic oscillations depends on the finding that direct ablation of other classes of ventral cord motor neurons (A, D, VC) does not. Could this question be addressed in part by ablating all B-neurons with acr-5::mini::SOG? Similarly, the unc-55 promoter is selectively expressed in VD and AS neurons and could therefore be used to determine if AS neurons are required since DD + VD function is not. The unc-53 promoter, expressed in DA + AS neurons, could also be used for this experiment since DA neurons are not required. The authors should consider these new experiments, which may be quite manageable in a short time, or provide a strong rationale for why they are not appropriate.

We thank the reviewer for these excellent suggestions. The conclusion that B/AS motor neurons are involved in rhythm generation does not rely entirely on the results of ablating other neuron types, but is also supported by two additional findings. First, we showed that optogenetic stimulation of B and AS in a tail paralyzed worm induces high frequency local undulations similar to 2FU (Figure 3—figure supplement 1 in the revised manuscript). Second, we showed that rhythmic hyperpolarization of the B motor neurons (P*acr-5::Arch*) suffices to reset and entrain the whole body undulatory rhythm (Figure 6—figure supplement 2). These results are consistent with the hypothesis that the B motor neurons are essential for rhythm generation.

As suggested, we have conducted further experiments to investigate the requirement of the B-type motor neurons in posterior rhythm generation.

We generated a strain incorporating *Punc-17beta::PH::miniSOG*, which was reported to specifically kill the A and B motor neurons. We found, however, that in our hands it preferentially kills DA and DB over VA and VB. We therefore combined *Punc17beta::PH::miniSOG* with our infrared laser ablation method to kill all B type motor neurons.

We found that ablation of DB and VB together resulted in tail paralysis and virtually eliminated optogenetically induced 2-frequency undulation (2FU) and undulations posterior to a nerve cord cut. Ablation of the DB motor neurons alone sharply reduced, but did not eliminate, incidence optogenetically induced 2-frequency undulation (2FU) and undulations posterior to a nerve cord cut. (Figure 5 in the revised manuscript).

In addition, we studied *vab-7* mutant worms, in which the DB motor neuron processes are aberrantly directed toward the head (opposite the normal direction). These worms exhibited a paralyzed tail and did not generate the high frequency undulation in 2FU (Figure 4—figure supplement 1 in the revised manuscript). The results further support the claim that B motor neurons are essential for rhythm generation.

With regard to AS, our experiments do not rule out a role or requirement for AS in generating the high frequency tail rhythms. We decided not to delve into this previously neglected class of motor neurons because we are aware that another group has undertaken a detailed study of AS. We leave open the possibility that AS has a role in rhythm generation.

All the results described above have been incorporated into the revised manuscript. In addition, a summary of the arguments in support of the B and/or AS neurons’ involvement in rhythm generation is now included in the Discussion.

4) Experiments described here either silence potential local sources of oscillation or sever physical connections between motor neurons and distal potential oscillators to conclude that motor circuit oscillations likely originate from distributed foci in the axial nerve cord. These experiments do not rule out the possibility, however, of "wireless" regulators potentially involving neuropeptide signals. The authors should address this point in the Discussion.

We have added a discussion of the possibility that non-synaptic regulators, transmitted across severed connections, may play a role in rhythm generation.

5) Subsection “Premotor interneurons are not essential for forward movement or 2FU”, last paragraph uses genetic mutants to eliminate gap junctions composed of the innexins UNC-7 and UNC-9 between AVB and B-type motor neurons to argue that the loss of AVB to B-type motor neuron electrical synapses does not prevent independent oscillations of anterior vs posterior body regions. This result is over-interpreted because unc-7 and unc-9 mutants would not eliminate other potential AVB B-motor neuron gap junctions composed of other innexins.

We agree that the original manuscript may have over-interpreted the results of the *unc7/unc-9* mutant experiments. We have revised the Discussion accordingly.

To clarify whether any AVB—B or other connections are required for 2FU, we performed new experiments in which we ablated AVB. We found that 2FU occurred only very rarely when AVB was removed. This was somewhat surprising because undulations occur readily when the VNC (and hence AVB’s process) is severed. However, we noted that AVB’s process disappeared when the cell body was ablated at the L4 stage, but when we severed the nerve cords directly the process was likely still present, and possibly functional, in each isolated circuit segment. This difference in residual AVB function may explain the observed behavioral differences. Although high frequency oscillations were rare after complete ablation of AVB, they were still occasionally possible, suggesting that AVB is important for but not strictly required for generation of these rhythms. These results were broadly consistent with AVB ablation experiments described in the preprint by Xu et al. These results are now included in the manuscript In Figure 4—figure supplement 1.